# Better Language Models Exhibit Higher Visual Alignment

**Jona Ruthardt**                                                      *jona.ruthardt@utn.de*
*Fundamental AI Lab, University of Technology Nuremberg*

**Gertjan J. Burghouts**                                         *gertjan.burghouts@tno.nl*
*Intelligent Imaging, TNO*

**Serge Belongie**                                                      *s.belongie@di.ku.dk*
*Department of Computer Science, University of Copenhagen*

**Yuki M. Asano**                                                      *yuki.asano@utn.de*
*Fundamental AI Lab, University of Technology Nuremberg*

**Reviewed on OpenReview:** *https: // openreview. net/ forum? id= wqBHJNqeQJ*
**Project Page:** *https: // jonaruthardt. github. io/ project/ ShareLock/*

## Abstract

How well do text-only large language models (LLMs) align with the visual world? We present a systematic evaluation of this question by incorporating frozen representations of various language models into a discriminative vision-language framework and measuring zero-shot generalization to novel concepts. We find that decoder-based models exhibit stronger visual alignment than encoders, even when controlling for model and dataset size. Moreover, language modeling performance correlates with visual generalization, suggesting that advances in unimodal LLMs can simultaneously improve vision models. Leveraging these insights, we propose *ShareLock*, a lightweight method for fusing frozen vision and language backbones. *ShareLock* achieves robust performance across tasks while drastically reducing the need for paired data and compute. With just 563k image-caption pairs and under one GPU-hour of training, it reaches 51% accuracy on ImageNet. In cross-lingual settings, *ShareLock* dramatically outperforms CLIP, achieving 38.7% top-1 accuracy on Chinese image classification versus CLIP's 1.4%. Code is available.

## 1 Introduction

Large Language Models (LLMs) are solely pretrained on unimodal textual data, yet are increasingly incorporated into systems that perceive and interact with the natural world (Ahn et al., 2022; Driess et al., 2023; Wayve, 2023). The lack of direct sensory experience raises fundamental questions as to what extent such models generalize across modalities and develop a meaningful understanding of *visual* reality. Do these models merely regurgitate visually relevant factual knowledge from their training corpus, or do they form internal representations that correspond to real-world phenomena? Despite their successful integration into large-scale Vision-Language Models (VLMs), judging the visual capabilities already inherent to LLMs is difficult. This stems not only from differences in training recipes and proprietary data, but especially from fine-tuning with *paired* image–text data, which blends with the visual knowledge already embedded in text-only models.

In contrast, Sharma et al. (2024) and Huh et al. (2024) more immediately assess the visual nature of LLMs and highlight a non-trivial degree of visual understanding and cross-modal alignment. These works compile proxy tasks such as generating code to represent real-world concepts (Sharma et al., 2024) or correlating vision and language features (Huh et al., 2024). However, reliance on highly constrained and synthetic tasks with limited practical significance fails to gauge the aptitude of LLMs in more realistic settings.

To this end, we assess visual alignment—the degree to which language model representations structurally and semantically correspond to those of vision models—through the task of zero-shot open-vocabulary image classification, as popularized by CLIP (Radford et al., 2021). This involves learning a projection of language embeddings into the vision manifold and selecting the label whose representation is most similar to a given image. To ensure a rigorous evaluation of *true* zero-shot generalization, we enforce strict disjointness between concepts encountered during training and testing as illustrated in Figure 2 (center) (Lampert et al., 2009). This mitigates concept leakage, a common issue in VLMs (Fig. 2, left). For example, Xu et al. (2024) find

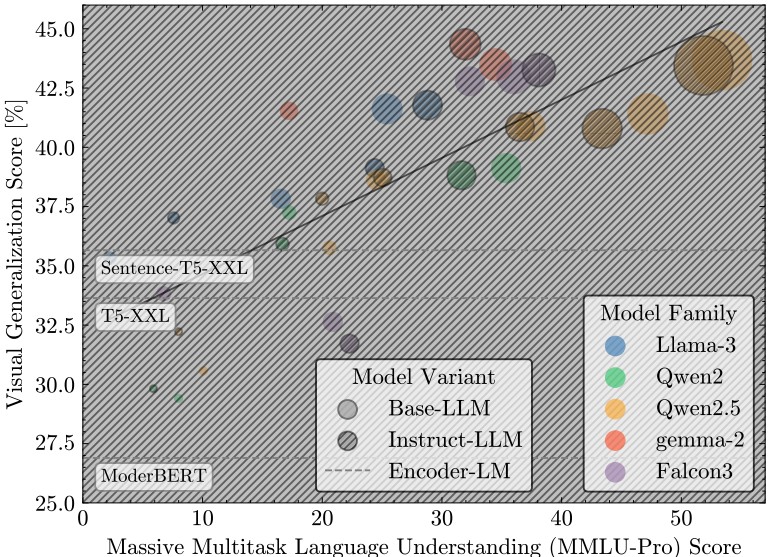

Figure 1: **Visual generalization *vs* language comprehension.** Language modeling capability on MMLU-Pro predicts LLM visual transfer performance (Pearson-$r$: 0.768). We compute a visual generalization score by aligning language with vision features in a CLIP-like framework and evaluating on disjoint sets of unaligned classes across four datasets. Dot size is proportional to the LLM's parameter count.

significant conceptual overlap between CLIP's training and evaluation datasets. Other works demonstrate sharp performance drops for less-frequent or truly novel concepts (Fang et al., 2022; Udandarao et al., 2024; Parashar et al., 2024; Mayilvahanan et al., 2025). In our proposed setup, generalization relies solely on the semantic information and visual knowledge encoded in language representations. By probing how well models capture visual semantics, we provide insight into their ability to encode text for vision-language applications. We observe non-trivial zero-shot generalization across language model types, indicating latent visual-semantic alignment. In particular, features from modern generative LLMs outperform classic encoder-based embeddings such as BERT (Devlin et al., 2019). This advantage of decoder-based models persists when controlling for pretraining data and parameter count across architectures. Intriguingly, we find that general LLM capability, as measured by MMLU-Pro (Hendrycks et al., 2021a), correlates positively with the model's visual performance, as shown in Figure 1. Even off-the-shelf LLMs without embedding-specific fine-tuning exhibit strong visual representation abilities.

Finally, we integrate frozen vision and language representations into a lightweight discriminative VLM, *ShareLock*, demonstrating strong multimodal capabilities across a range of tasks. Despite using only a fraction of the data and learnable parameters, our method approaches the performance of fully optimized CLIP models trained on orders of magnitude more paired data. By capitalizing on the broad pretraining of modern LLMs, *ShareLock* achieves remarkable cross-lingual zero-shot generalization to non-English languages, outperforming CLIP's performance of 1.4% with 38.7% for Chinese. The visual and linguistic expressiveness of LLM representations is particularly effective in nuanced and fine-grained tasks, resulting in above-CLIP compositional reasoning performance. Our results highlight the considerable extent to which language representations capture visual structure and semantics, enabling highly efficient alignment with vision embeddings using limited supervision and parameterization.

Overall, the main contributions of this work are:

- We provide a systematic evaluation of the visual alignment inherent to language models, using strict zero-shot image classification as a practically-relevant probing task.

- Our analysis highlights modern decoder-based LLMs as effective sources of visual knowledge, with semantically meaningful representations extractable from their internal states.

- With *ShareLock*, we incorporate frozen LLMs with high intrinsic visual alignment into a lightweight VLM, resulting in improved robustness and generalization on various classification, retrieval, multilingual understanding, and compositional reasoning tasks.

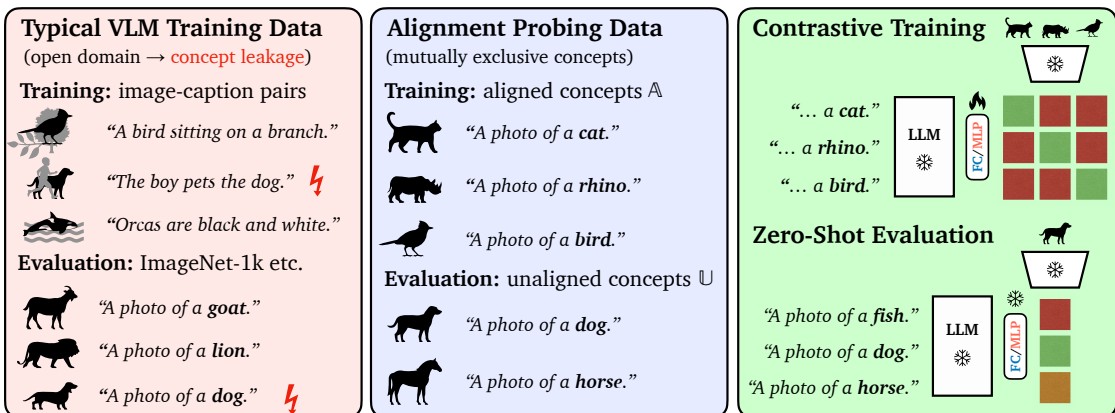

Figure 2: **Comparison of training modes.** *(Left)* Web-scale VLMs training lacks concept control, weakening generalization claims and resulting in erratic drops for rare categories. *(Center)* Our visual alignment probing protocol enforces strict concept separation to assess *true* generalization. *(Right)* Our ShareLock method uses lightweight projections to align frozen unimodal models via CLIP-style contrastive learning and zero-shot evaluation.

## 2  Related Work

**Visual Understanding of Large Language Models.**   LLMs can infer and reason about visual content without explicit multi-modal training (Bowman, 2023). Sharma et al. (2024) tasked LLMs to draw common objects and scenes using simple shapes, indicating spatial understanding and illustrating that LLMs can conceptualize real-world settings. Various works highlight the plausibility and utility of LLM-generated descriptions of objects in the context of image classification and demonstrate that LLMs possess encyclopedic knowledge about visual characteristics (Pratt et al., 2022; Menon and Vondrick, 2023; Yang et al., 2022; Saha et al., 2024). These capabilities suggest that the extensive pretraining on large volumes of diverse textual data aids the visual understanding of LLMs. Huh et al. (2024) argue that the embedding spaces of neural networks converge towards a shared 'platonic' representation of reality irrespective of the concrete optimization objectives and data modality utilized during training. Similarly, we investigate the degree of visual alignment inherent to exclusively language-based representations but assess this in the practically more relevant context of zero-shot image classification and design a rigorous benchmark to measure the true generalization capabilities facilitated by such language embeddings.

**Vision-Language Alignment.**   Similarly to Huh et al. (2024), other recent works indicate and exploit alignment between pretrained unimodal vision and language models. Zhai et al. (2022) and Khan and Fu (2023) reveal that leveraging pretrained models and only tuning a subset of parameters can improve performance and efficiency over CLIP (Radford et al., 2021), indicating some degree of model-inherent alignment. Norelli et al. (2023) and Maniparambil et al. (2024) only rely on cross-modal correlations for training-free alignment of unimodal models. However, these studies are primarily restricted to encoder-based language models. Zhang et al. (2024) incorporate decoder-based LLMs into a CLIP-like framework but transform both vision and language representations, making it difficult to isolate the contribution of each modality during alignment. In contrast, we systematically evaluate both encoder- and decoder-based language models, examine how well their representations can be mapped to visual latent spaces, and focus on zero-shot generalization.

## 3  Probing LLMs for Visual Alignment

This section details our methodology for quantitatively assessing the visual alignment of language models. We introduce the zero-shot evaluation protocol, describe the architecture aligning frozen vision and language backbones, and summarize implementation details to ensure reproducibility.

## 3.1 Zero-Shot Generalization

We determine the visual aptitude of language models by drawing on traditional zero-shot learning methodology (cf. Lampert et al. (2009)) and consider how their representations facilitate generalization to novel concepts. To rigorously assess *true* generalization performance without concept leakage from supervision with arbitrary web-scraped image-captions pairs, we split image classification datasets into *aligned* classes $\mathbb{A}$ (training stage) and *unaligned* classes $\mathbb{U}$ (testing stage) with $\mathbb{A} \cap \mathbb{U} = \emptyset$ (see Figure 2, center). In the absence of image-specific captions, text-based class representations $\boldsymbol{f}(y_i)$ are used as supervision signals during training and for zero-shot transfer during inference. Crucially, this implies that the performance to discriminate novel concepts is contingent on the validity and cross-modal continuity of the class representations. Therefore, this setup allows us to assess the degree to which language models encode visual knowledge and semantics.

## 3.2 Shared Vision-Language-Locked Tuning

To map textual inputs into visual latent spaces, we draw inspiration from late-fusion architectures in CLIP-like models. Texts are first encoded using a language model $\phi_{\text{txt}}(\cdot)$ and subsequently projected into the $d$-dimensional latent space of the vision encoder $\phi_{\text{img}}(\cdot)$ via a learnable projection network $\mathbf{p}_{\text{txt}}(\cdot)$. The latent representation for a given input image $\mathbf{x}_i$ or caption $\mathbf{t}_i$ is therefore computed by $\mathbf{z}_{\text{img}} = \phi_{\text{img}}(\mathbf{x}_i) \in \mathbb{R}^d$ and $\mathbf{z}_{\text{txt}} = \mathbf{p}_{\text{txt}}(\phi_{\text{txt}}(\mathbf{t}_i)) \in \mathbb{R}^d$, respectively. Their similarity $\text{sim}(\mathbf{z}_{\text{img}}, \mathbf{z}_{\text{txt}})$ is computed as the cosine similarity, given by the dot product of the normalized embeddings.

During training, only the lightweight projection network $\mathbf{p}_{\text{txt}}(\cdot)$ is optimized, while the pretrained vision and language backbones remain frozen. A contrastive loss encourages alignment by pulling textual representations closer to their corresponding image embeddings while pushing them away from non-matching ones, as in (Radford et al., 2021). For an image-text pair $i$ in a batch with $N$ items, it is given by

$$\mathcal{L}(i) = -\log \frac{\exp\left(\text{sim}(\mathbf{z}_{\text{m}}^i, \mathbf{z}_{\text{n}}^i)/\tau\right)}{\sum_{j=1}^{N} \exp(\text{sim}(\mathbf{z}_{\text{m}}^i, \mathbf{z}_{\text{n}}^j)/\tau)}, \tag{1}$$

for both alternated modalities pairings $(m, n) \in \{(\text{txt}, \text{img}), (\text{img}, \text{txt})\}$ and with $\tau$ being a fixed temperature parameter. Given a set of classes $\mathbb{C}$ and their corresponding textual class representations $\boldsymbol{f}(\cdot)$, the predicted class $\hat{c}$ for a sample $\mathbf{x}_i$ is obtained via

$$\hat{c} = \arg\max_{c \in \mathbb{C}} \text{sim}(\mathbf{z}_{\text{img}}, \mathbf{p}_{\text{txt}}(\phi_{\text{txt}}(\boldsymbol{f}(c)))). \tag{2}$$

## 3.3 Experimental Setup

**Datasets.** For a comprehensive evaluation, we select four datasets: AWA2 (Xian et al., 2017), CUB (Wah et al., 2011), FGVCAircraft (Maji et al., 2013), and ImageNet$^+$, spanning natural and human-made artifacts, coarse and fine-grained categories, and varying scales ($40 \leq |\mathbb{A}| \leq 1000$). ImageNet$^+$ treats ImageNet-1k (Deng et al., 2009) classes as aligned concepts and the 500 most populated ImageNet-21k classes as unaligned ones. For AWA2 and CUB, we use splits by Xian et al. (2017) while randomly dividing aircraft types into 50 aligned and 20 unaligned classes. We report the average per-class classification accuracy on unaligned classes $c\mathbb{A}$ across the datasets as a measure of visual generalization ability facilitated by the language embeddings.

Besides the class-name-based templates proposed by Radford et al. (2021) (e.g., `"a photo of a <class>"`), we generate more comprehensive Wikipedia-style descriptions with an LLM and acquire human-curated information from Wikipedia to be used as class representations (details in C).

**Pretrained Unimodal Backbones.** Given its strong performance, broad pretraining regime, and popularity, the ViT-L/14 variant of the self-supervised DINOv2 model family (Oquab et al., 2023) is the default vision backbone. Global image embeddings are obtained through the `CLS` token. Language representations are extracted from encoder-based models through mean token pooling or directly via the `CLS` token if fine-tuned on sentence-level representation tasks. For decoder-based LLMs other than NV-Embed (Lee et al., 2024), features are extracted through last-token pooling (details in Appendix). Frozen vision and language model features are initially precomputed and stored for direct re-use in subsequent epochs.

**Training.** Given the data-constrained setting, we optimize a linear projection network using Adam (Kingma and Ba, 2014) with a learning rate following a cosine schedule with a maximum value of $10^{-3}$. Gradient clipping to a global norm of 1 and weight decay of $10^{-4}$ are applied. The loss of Eqn. 1 is applied with

Table 1: **Visual generalization scores of various language models.** We benchmark visual alignment of models for different class representations. Decoder-based language models outperform popular encoder-based architectures across all types of input data. Llama-3 8B (Instr.) is used for LLM generated Wikipedia articles.

| | Language Model [# Parameters] | Class Names | LLM Wikip. | original Wikip. |
|---|---|---|---|---|
| Enc. | BERT-Large [336M] (Devlin et al., 2019) | 14.2 | 16.4 | 24.0 |
| | ModernBERT [395M] (Nussbaum et al., 2024) | 28.9 | 23.9 | 24.8 |
| | all-roberta-v1 [355M] (Liu et al., 2019) | 31.0 | 34.5 | 41.7 |
| | SentenceT5-XL [1B] (Ni et al., 2022) | 32.8 | 36.1 | 39.6 |
| | SentenceT5-XXL [5B] (Ni et al., 2022) | 36.6 | 39.1 | 42.8 |
| | Flan-UL2 [10B] (Tay, 2024) | 37.9 | 41.0 | 44.0 |
| Dec. | NV-Embed-v2 [8B] (Lee et al., 2024) | 39.4 | 42.1 | 43.6 |
| | Llama-3 [8B] (Dubey et al., 2024) | 39.8 | 43.9 | **44.5** |
| | Gemma-2 [9B] (Mesnard et al., 2024) | **42.8** | **45.0** | **44.5** |

$\tau = 0.07$, and models are trained until convergence on a randomly chosen validation split or for a maximum of 3.5k steps with a batch size of 16,384 and five different initialization seeds. Dropout (Srivastava et al., 2014) with $p = 0.2$ is applied during training.

## 4 Language-driven Visual Generalization

Utilizing the zero-shot evaluation methodology outlined, we investigate critical factors that promote the generalization of language representations to illuminate language models' visual capabilities.

**LLM representations encode visual knowledge.** The class-wise supervision and limited concept diversity of conventional image classification datasets can impede vision-language alignment but permit more sophisticated semantic class representations beyond simple template-based targets as typically used with CLIP-like models (Radford et al., 2021; LAION AI, 2022). We thus examine how the nature and information content of different textual class representations[2] impact generalization performance. The results are summarized in Table 1. As a naive language-free baseline, we also construct one-hot-encoded class representations. Unsurprisingly, the lack of semantic continuity connecting aligned and unaligned concepts results in a near-random generalization score of 4.5% and demonstrates that generalization in the proposed evaluation protocol is fundamentally dependent on the semantic alignment of vision and language embeddings. We find that the addition of auxiliary information, such as Wikipedia articles, results in improved performance for most language models, reflecting insights from previous studies (Pratt et al., 2022; Menon and Vondrick, 2023; Yang et al., 2022; Saha et al., 2024). We also find that LLM-generated articles describing a class in the style of Wikipedia ("LLM Wikip." in Table 1) can provide strong targets during multi-modal alignment, achieving the best overall performance of 45.0%. Interestingly, relying on strictly human-curated data in the form of actual Wikipedia articles tends only to provide marginal benefits, for example, from 43.9% → 44.5% and 42.1% → 43.6% for Llama-3 and NV-Embed. Thus, LLMs can effectively absorb and interpolate substantial amounts of factual information from their training data, positioning them as valuable sources of visually relevant knowledge.

**Decoders excel in visual concept representation.** A new insight resulting from our analysis is the competitiveness of decoder-based language models in representing visual concepts. Compared to language encoders commonly used in vision tasks, we find that representing inputs with decoders can result in higher performance, mirroring a recently emerging trend in the language domain (Lee et al., 2024; Springer et al., 2024). However, results in Table 1 confound factors such as model and dataset size with the different pretraining tasks. To isolate the impact of architecture, we conducted controlled comparisons with the Ettin model family (Weller et al., 2025) that only varies attention patterns and training objectives while unifying data and parameter count for encoder- and decoder-based models. Figure 3 reveals that decoders outperform encoders by an average of 2.7 percentage points across model sizes, suggesting that autoregressive next-token prediction gives rise to more vision-aligned representations compared to masked language modeling

---

[2]Details about the characteristics and acquisition of these class representations are elaborated in Appendix C

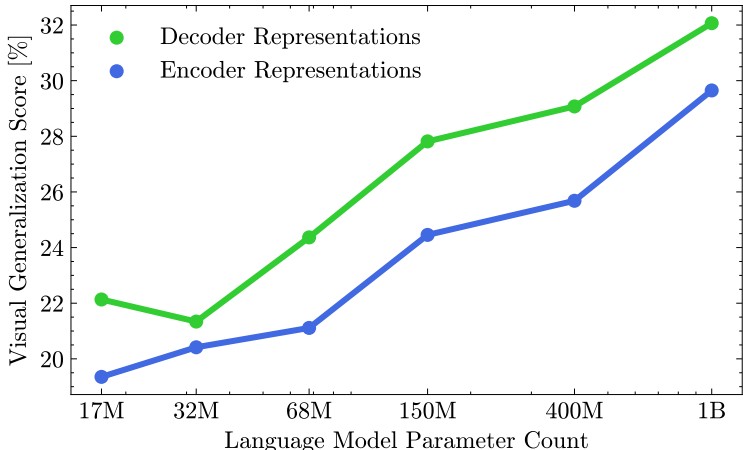

Figure 3: **Encoder- *vs* decoder-based language models.** The models are trained on identical data with matched model size, isolating the effects of their pretraining objectives. Decoders demonstrate higher visual alignment across all model sizes compared to encoder models.

in encoder models. Beyond this intrinsic advantage, decoder models are often pretrained on larger-scale data and comprise more parameters, leading to more capable models in addition to architectural benefits. For instance, Gemma-2 9B achieves the highest score of 45.0% across our evaluated input types, outperforming the strongest and similarly sized encoder model, Flan-UL2, which reaches 41.0% on the same input data. Other decoders like LLaMA and NV-Embed perform on par with Gemma, whereas encoder models such as T5- and BERT-based variants lag considerably behind. These findings suggest that decoder-based LLMs can offer both architectural and practical advantages for visual representation and alignment tasks.

**LLM and its visual performance are correlated.** In Figure 1, we compare various LLMs by the visual generalization ability they possess, as well as their MMLU-Pro (Wang et al., 2024a) score taken from the Open LLM Leaderboard (Fourrier et al., 2024), a common metric to measure LLM performance. We find that the general capability of language models is strongly correlated with their ability to perform well on the visual tasks (Pearson coefficient $r$: 0.768). Within and across model families, we see improved visual generalization as the capacity and capabilities of models increase. Since models steadily improve in the language domain, our evaluation protocol will be helpful in assessing whether the trend of increasing visual understanding will continue in future LLM models. If this holds, VLMs that incorporate LLMs can piggyback off developments in the language modeling domain.

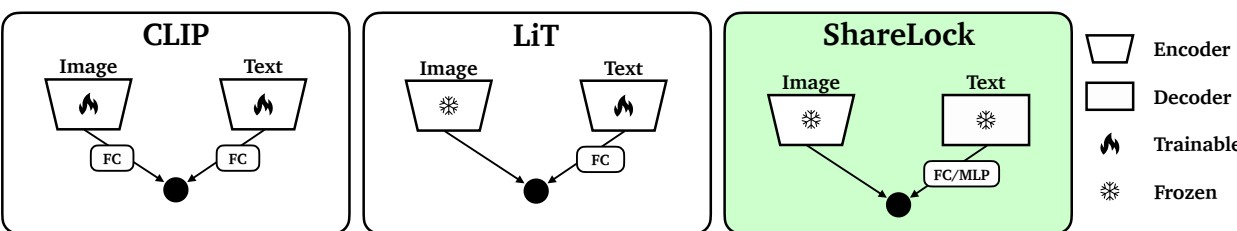

Figure 4: **Our *ShareLock vs* previous methods.** Compared to CLIP and LiT, *ShareLock* utilizes frozen pretrained representations for both modalities, allowing extremely efficient training. Using this framework, we assess how "visual" frozen language models' text representations are by how strong the resulting model can generalize to entirely novel categories. We find decoder-only LLMs to yield strong performances for zero-shot generalization and hence incorporate them as the text backbones.

# 5    LLMs for General-Purpose VLMs

The previous section demonstrated that LLMs trained for textual next-token prediction encode visually-aligned semantics in their representations. Next, we investigate whether this inherent visual alignment can be leveraged to obtain effective and efficient multimodal models by integrating off-the-shelf LLMs directly into CLIP-like VLMs. For this, we relax the strict zero-shot setup and utilize larger-scale image-caption datasets (see Figure 2, left) to explore the benefits and limitations of general-purpose VLMs that fuse existing frozen foundation models with minimal data and compute.

## 5.1    Experimental Setup

**Methodology.**    We propose integrating LLMs into a CLIP-like framework through "**Share**d Vision-Language-**Lock**ed Tuning" (*ShareLock*), extending the architecture and training setup introduced in Section 3. As illustrated in Figure 4, this design places a LLM at the core of the model and maps its expressive representations into the feature space of a frozen vision model. If not noted otherwise, Llama-3 8B (Dubey et al., 2024) and DINOv2-L (Oquab et al., 2023) are used as frozen backbones. Given the greater availability of training data compared to the strict zero-shot setting, we expand the projection network $\mathbf{p}_{\text{txt}}(\cdot)$ applied on top of the frozen language features to four layers with a hidden dimension of 4096. The additional parameters, combined with ReLU activations between layers, enhance *ShareLock*'s ability to capture complex, non-linear cross-modal correspondences. Additionally, the maximum number of optimization steps is increased to 5000.

**Datasets.**    Our investigation focuses on leveraging LLMs with minimal additional paired data and explores how unimodal embeddings can drive robust multimodal performance with minimal supervision and alignment. As a result, our evaluation is limited to comparably small paired datasets. **COCO Captions.** COCO Captions (Chen et al., 2015) contains 83k images with multiple human-written captions per image, from which we randomly sample during training. **CC3M.** Conceptual Captions (Sharma et al., 2018) comprises 2.8M filtered web-scraped image-alt-text pairs. We also utilize a smaller subset with more balanced concept coverage designed for LLaVA (Liu et al., 2023). **CC12M.** CC12M (Changpinyo et al., 2021) expands CC3M by 12M image-text pairs, enabling evaluation at larger scales. Due to expired links, our version contains about 8.5M samples. If not noted otherwise, CC3M is used to compare design choices.

**Computational efficiency and storage.**    Precomputing features for the CC3M-Llava subset (563k pairs) using Llama-3 8B requires around 8 hours on a single A100 40GB GPU. Extracting DINOv2 features and optimizing the MLP-based projection network take approximately 1 GPU hour each, totaling roughly 10 GPU hours. This is paired with a significant storage reduction from over 80GB for the raw data to just 12GB for precomputed features. For classification at inference, language targets are computed once per label set and inference cost is dominated by the vision encoder. Consequently, CLIP, LiT, and ShareLock have comparable FLOPs and latency when using the same backbone architecture (e.g., ViT-L/14@224x224: $\approx$ 160 GFLOPs), despite ShareLock's larger LLM.

**Evaluation.**    We employ a comprehensive suite of evaluations to assess *ShareLock*'s capabilities across a wide range of tasks. Based on the publicly available CLIP Benchmark (LAION AI, 2022), we gauge the models' zero-shot classification abilities across diverse datasets: ImageNet-1k  (Deng et al., 2009), ImageNet-R(endition) (Hendrycks et al., 2021b), ImageNet-A(dversarial) (Hendrycks et al., 2021c), ImageNet-S(ketch) (Wang et al., 2019), Oxford-IIIT Pets (Parkhi et al., 2012), Oxford Flowers (Nilsback and Zisserman, 2008), Stanford Cars (Krause et al., 2013), and FGVCAircraft (Maji et al., 2013). We also provide qualitative text-to-image retrieval results on ImageNet for CC3M-trained models. Moreover, the challenging compositionality Winoground task (Thrush et al., 2022) is explored.

**Baselines and comparisons.**    We compare our straightforward and economical way of incorporating LLMs into VLMs to more conventional CLIP-like methods, particularly emphasizing data-efficient alignment approaches. Alongside the original ViT-B/16 variant of CLIP (Radford et al., 2021), we test against several CLIP-like models trained on public datasets of different scales and with modified learning objectives that all share the standard dual-encoder architecture where both image and text encoders are trained from scratch with a contrastive loss (Fig. 4, left) (Fan et al., 2023; Gadre et al., 2023; Mu et al., 2022). Leveraging pretrained models, we evaluate how *ShareLock* compares to LiT (Zhai et al., 2022) and ASIF (Norelli et al., 2023) by reproducing these methods on smaller-scale datasets. LiT corresponds to the partially frozen setup in Figure 4 (center), where the vision encoder is locked but the text tower and projection are fine-tuned.

Table 2: **Open-vocabulary zero-shot classification accuracy on various datasets.** Especially in low-data regimes, the frozen LLM features (Llama-3 8B) utilized by *ShareLock* enable it to outperform CLIP, LiT and ASIF baselines and achieves performances competitive with models trained on significantly more paired data, such as CommonPool-L (384M).

| Model | Dataset | Size | IN-1k | IN-R | IN-A | IN-S | Pets | Flowers | Cars | Aircraft | Avg |
|---|---|---|---|---|---|---|---|---|---|---|---|
| LiT | COCO | 83k | 21.4 | 35.5 | 21.4 | 18.9 | 23.5 | 7.0 | 2.0 | 2.1 | 16.5 |
| ASIF | COCO | 83k | 9.4 | 14.4 | 8.8 | 6.9 | 7.0 | 1.6 | 1.3 | 2.8 | 6.5 |
| *ShareLock* | COCO | 83k | **36.9** | **49.0** | **37.0** | **29.8** | **34.5** | **10.5** | **4.4** | **7.9** | **26.2** |
| LiT | CC3M Subset | 563k | 44.5 | 70.0 | 58.3 | 39.5 | 25.5 | 34.4 | 2.5 | 2.4 | 34.6 |
| ASIF | CC3M Subset | 563k | 21.6 | 27.7 | 24.4 | 14.9 | 11.7 | 6.4 | 2.3 | 2.1 | 13.9 |
| *ShareLock* | CC3M Subset | 563k | **51.5** | **71.9** | **63.6** | **43.2** | **33.0** | **39.2** | **5.1** | **6.5** | **39.2** |
| CLIP | CC3M | 2.8M | 16.0 | 17.6 | 3.6 | 6.4 | 13.0 | 10.8 | 0.8 | 1.4 | 8.7 |
| SLIP | CC3M | 2.8M | 23.5 | 26.8 | 6.8 | 12.1 | 17.0 | 13.5 | 1.2 | 1.3 | 12.8 |
| LaCLIP | CC3M | 2.8M | 21.3 | 23.5 | 5.0 | 10.6 | 15.8 | 15.7 | 1.6 | 1.6 | 11.9 |
| LiT | CC3M | 2.8M | 46.8 | 72.8 | 59.4 | 40.8 | 31.1 | **42.4** | 3.7 | 2.6 | 37.4 |
| *ShareLock* | CC3M | 2.8M | **54.5** | **74.7** | **65.9** | **46.0** | **36.0** | 38.9 | **7.5** | **6.7** | **41.3** |
| DataComp | CPool-S | 3.84M | 3.0 | 4.4 | 1.5 | 1.3 | 4.0 | 1.8 | 1.6 | 1.4 | 2.4 |
| CLIP | CC12M | 12M | 41.6 | 52.6 | 10.7 | 28.8 | 64.2 | 36.7 | 24.1 | 2.5 | 32.6 |
| SLIP | CC12M | 12M | 41.7 | 55.2 | 13.8 | 30.7 | 56.7 | 34.1 | 22.4 | 3.0 | 32.2 |
| LaCLIP | CC12M | 12M | 49.0 | 63.8 | 14.7 | 39.4 | 72.5 | 43.2 | **36.2** | 5.5 | 40.5 |
| LiT | CC12M | 8.5M | 59.9 | **79.9** | 68.2 | 50.6 | **76.8** | 51.9 | 13.5 | 6.0 | 50.8 |
| *ShareLock* | CC12M | 8.5M | **62.0** | 78.5 | **70.1** | **51.6** | 71.3 | **56.3** | 15.0 | **10.9** | **52.0** |
| DataComp | CPool-M | 38.4M | 23.0 | 28.0 | 4.3 | 15.1 | 29.9 | 22.4 | 22.0 | 1.7 | 18.3 |
| DataComp | CPool-L | 384M | 55.3 | 65.0 | 20.2 | 43.2 | 77.8 | 53.3 | 67.7 | 7.1 | 48.7 |
| CLIP | Proprietary | 400M | 68.4 | 77.6 | 50.1 | 48.2 | 89.0 | 71.2 | 64.7 | 24.4 | 61.7 |

*ShareLock* follows the fully frozen setup in Figure 4 (right), freezing both vision and language encoders and learning only lightweight projection networks. For LiT baselines, we initialize the language encoder with pretrained BERT-Base weights (Devlin et al., 2019), following Zhai et al. (2022). When comparing *ShareLock* with LiT and ASIF, the same precomputed features (except for LiT's language inputs) are used.

## 5.2 Comparison to Conventional VLMs

**Classification.** LLMs can directly be leveraged profitably in vision-centric tasks and outperform conventional models trained on similar small-scale datasets as demonstrated by the IN-1k accuracies in Table 2. *ShareLock*'s 54.5% accuracy on CC3M substantially exceeds both CLIP (16.0%) and LiT (46.8%), despite the latter using the same vision features and fully fine-tuning the language component. Even with larger datasets like CC12M, where full fine-tuning becomes more viable, minimal transformations on top of LLM representations maintain a competitive advantage of $3\% - 15\%$ over LiT and CLIP. Compared to the training-free ASIF, optimizing a small number of parameters proves advantageous and forgoes the reliance on large and diverse reference datasets and extensive compute during inference.

Beyond competitive performance on general-purpose classification, leveraging strong representations from pretrained models enables increased robustness to out-of-distribution image inputs as seen in columns "IN-R" to "IN-A" of Table 2, surpassing the robustness of the original CLIP model despite being exposed to a fraction of the training data (8.5M vs. 400M).

The fine-grained nature of certain classification problems (cols. "Pet" to "Aircraft" in Tab. 2) demands larger-scale datasets with more diverse and nuanced concepts included as minute visual differences may be insufficiently captured in the text space. Consequently, low performance on small datasets can be observed across all methods. Nonetheless, the LLM representations still contain visually valuable signals, enabling *ShareLock* to surpass other methods trained on the same data in 15/16 cases and demonstrating effective utilization of intrinsic LLM knowledge to generalize to truly novel concepts.

**Multilingual Understanding.** Most popular multimodal datasets consist primarily of English captions. Consequently, VLMs like CLIP and LiT experience significant performance drops when performing inference in other languages, as shown in Table 3 on multilingual ImageNet LAION AI (2022). In contrast, the broader and typically multilingual pretraining of LLMs enables *ShareLock* to harness cross-linguistic image-text

Table 3: **Multilingual zero-shot classification accuracy.** Leveraging extensive pretraining and consistent representations, *ShareLock* allows cross-lingual transfer on ImageNet without extra alignment.

| Model | Dataset | [Size] | EN | CN | JP | IT |
|---|---|---|---|---|---|---|
| LiT | COCO | 83k | 21.4 | 0.2 | 0.2 | 3.6 |
| *ShareLock* | COCO | 83k | **36.9** | **20.0** | **11.2** | **15.8** |
| CLIP | CC12M | 12M | 41.6 | 0.1 | 0.1 | 7.9 |
| LiT | CC12M | 8.5M | 59.9 | 0.2 | 0.1 | 12.9 |
| *ShareLock* | CC12M | 8.5M | **62.0** | **38.7** | **19.8** | **39.3** |
| DataComp | CPool-M | 38.4M | 23.0 | 0.2 | 0.3 | 4.7 |
| DataComp | CPool-L | 384M | 55.3 | 0.7 | 1.5 | 15.2 |
| CLIP | Proprietary | 400M | 68.4 | 1.4 | 4.1 | 21.7 |

Table 4: **Compositional reasoning on Winoground [Accuracy].** Strong frozen language features alone do not address systemic shortcomings inherent to contrastive alignment approaches when it comes to spatial or conceptual relationships, but enable *ShareLock* to outperform all alternative methods on image selection.

| Model | Dataset | [Size] | Text | Image | Group |
|---|---|---|---|---|---|
| Human | | | **89.5** | 88.5 | 85.5 |
| Chance | | | **25.0** | 25.0 | 16.7 |
| LiT | COCO | 83k | **21.3** | 7.3 | 3.5 |
| ASIF | COCO | 83k | 18.8 | 9.0 | 5.3 |
| *ShareLock* | COCO | 83k | 20.5 | **12.5** | **6.5** |
| CLIP | CC12M | 12M | 22.3 | 9.5 | 5.3 |
| LiT | CC12M | 8.5M | 22.0 | 6.5 | 4.0 |
| *ShareLock* | CC12M | 8.5M | **25.0** | **12.5** | **9.5** |
| DataComp | CPool-M | 38.4M | 25.0 | 8.3 | 6.3 |
| DataComp | CPool-L | 384M | 27.0 | 9.5 | 7.0 |
| CLIP | Propriatary | 400M | 30.8 | 10.8 | 8.3 |

consistencies, greatly mitigating the performance loss in non-English languages. Even with substantially fewer training samples, *ShareLock* surpasses the original CLIP model, achieving an accuracy of 38.7% versus 1.4% for Chinese and 19.8% versus 4.1% for Japanese. In comparison, both LiT and CLIP models similarly trained on CC12M demonstrate near-random performance in these languages. This transfer of capabilities makes the inclusion of LLMs especially attractive for low-resource languages with little available or high-quality multimodal data.

**Compositionality.** Late-fusion VLMs often struggle with nuanced textual and fine-grained compositional differences, as seen in benchmarks like Winoground (Thrush et al., 2022) and SugarCrepe (Hsieh et al., 2023). Despite the intriguing linguistic and generative abilities of LLMs, their representations fail to adequately reflect fine linguistic differences in the vision-language contrastive setting. While *ShareLock* improves image scores over CLIP (12.5 vs. 10.8) and thus more reliably selects the correct image given a textual description, it still falls short of significant above-random performance and remains far from human-level capability. However, the low performance on compositionality tasks might partly be an architectural limitation, as recent works (Zhang et al., 2024; Jose et al., 2024) have indicated limitations of solely aligning language representation to the vision space, as suggested by Zhai et al. (2022). Thus, also applying transformations on top of vision features can be beneficial, especially in retrieval and detail-oriented settings.

**Data scaling.** Figure 5 illustrates that *ShareLock* achieves comparable performance to CLIP and DataComp models while using orders of magnitude less data. Utilizing frozen LLM features is especially effective in low-data regimes, consistently outperforming conventional CLIP-like models, further underlining their visually-relevant semantic content and capacity to facilitate generalization. Additionally, tuning far fewer parameters enables substantially larger batch sizes (see Table 5), which has been shown to improve contrastive learning performance (Zhai et al., 2022). Frozen backbones also enables the integration of large-scale LLMs into CLIP-like architectures while maintaining training efficiency.

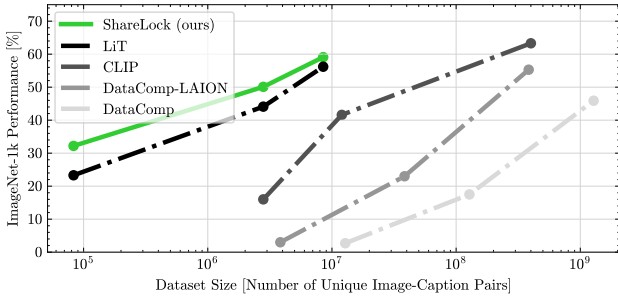

Figure 5: **Scaling of image-text dataset size.** *Share-Lock* outperforms other models despite using notably fewer datapoints.

Table 5: **Maximum batch size @80GB.** Frozen backbones permit *ShareLock* to utilize significantly larger batch sizes.

| Method | Language Backbone | |
| --- | --- | --- |
| | **BERT-base** | **Llama-3 (8B)** |
| CLIP | $1,600$ | 0 |
| LiT | $2,450$ | 0 |
| *ShareLock* | **61,000** | **57,000** |

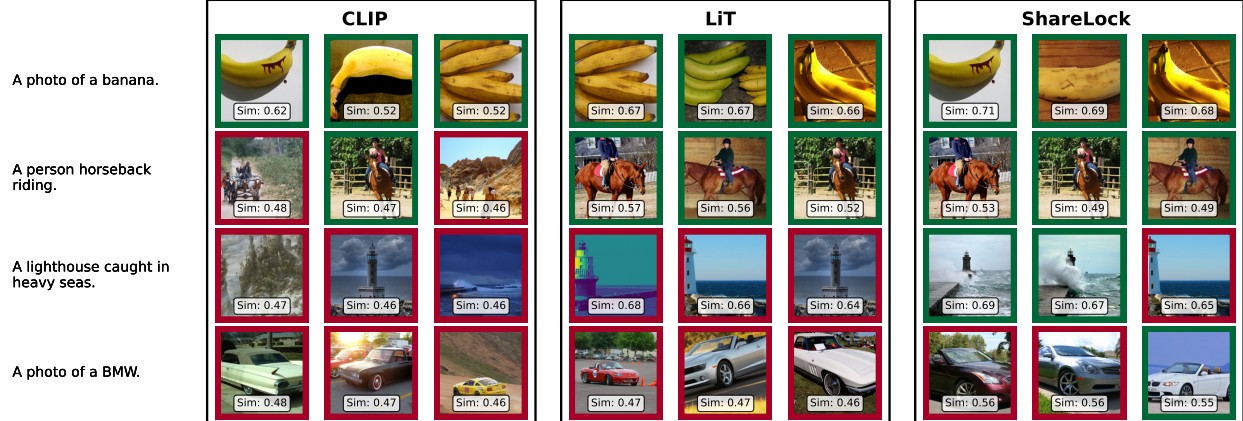

Figure 6: **Qualitative comparison.** We show qualitative top-3 retrieval results for CLIP, LiT and *ShareLock* models trained on CC3M. Green border color indicates correctly retrieved samples.

## 5.3 Qualitative Results

In addition to quantitative evaluations, Figure 6 demonstrates *ShareLock*'s strong text-image alignment across diverse prompts, showing advantages over CC3M-trained CLIP and LiT models for both fine-grained (e.g., *"a photo of a BMW"*) and abstract (e.g., *"[...] heavy seas"*) queries.

## 5.4 Component Analysis

**Choice of language model**  As the nature and quality of the frozen language features are of great significance in the proposed architecture, we examine the choice of language model on the CC3M dataset. Reflecting the insights from our investigation in Section 4, Table 6a highlights the potential of decoder-based models for vision-language tasks. Although BERT encoders serve as the starting point in LiT models, they perform poorly without fine-tuning. Similarly, all-roberta-v1 (Liu et al., 2019) improves significantly but remains inferior to LLM-based representations, despite being highlighted by Maniparambil et al. (2024) for its high inherent visual alignment. In contrast, frozen decoder-based representations consistently surpass BERT-based ones, with gains of 40% to 350%, showcasing the richness of strong LLM representations from the more extensive pretraining and larger parameter counts. Even naive last-token-pooling is suitable for extracting visual information from LLMs and the more sophisticated multi-token embedding approach of NV-Embed (Lee et al., 2024) fails to materialize in consistent and notable performance gains over off-the-shelf Llama and Gemma models.

**Choice of vision encoder**  As *ShareLock* is agnostic to the utilized vision encoder, we compare variants with differing architectures and supervision regimes in Table 6b. Since language embeddings map to the vision space, encoder choice is a crucial factor, as reflected in notable performance differences. Unlike DINOv2, the

Table 6: **Comparison of unimodal backbones.** Strong and comprehensive features rule generalization.

(a) **Comparison of language backbones.** Decoder-based LLMs facilitate generalization most. Popular encoders lag behind when not tuned.

| | Lang. Model | IN1k | IN-R | IN-A | Pets | Cars |
|---|---|---|---|---|---|---|
| Enc. | BERT-Base | 44.3 | 71.7 | 59.6 | 18.3 | 2.2 |
| | all-roberta-v1 | 49.5 | 72.7 | 61.5 | 27.4 | 4.3 |
| Dec. | Llama-3 8B | 54.5 | 74.7 | 65.9 | 36.0 | 7.5 |
| | Gemma-2 9B | 56.4 | **76.0** | **68.1** | **49.9** | 6.9 |
| | NV-Embed-v2 | **57.0** | 75.9 | 66.9 | 46.3 | **8.0** |

(b) **Comparison of vision backbones.** Self-supervised backbones transfer best across dataset and benefit from increased model capacity.

| | Vision Model | IN1k | IN-R | IN-A | Pets | Cars |
|---|---|---|---|---|---|---|
| Sup. | ResNet101 | 44.6 | 5.1 | 34.1 | 28.4 | 4.7 |
| | ConvNextV2-L | **64.7** | 60.6 | 54.9 | 28.8 | **8.4** |
| | ViT-L | 55.0 | 38.0 | 16.0 | 29.3 | 6.1 |
| Self-sup. | DINOv2-S | 45.6 | 49.7 | 27.8 | 33.3 | 5.4 |
| | -B | 52.1 | 64.1 | 50.9 | **43.1** | 4.4 |
| | -L | 54.5 | 74.7 | 65.9 | 36.0 | 7.5 |
| | -G | 56.3 | **77.7** | **69.9** | 36.2 | 6.6 |

other models were only trained on ImageNet and exhibit lower robustness and generality on other datasets, highlighting the benefits of broad pretraining across diverse concepts – even without explicit supervision. One advantage of CLIP is its favorable scaling characteristics when increasing the size of the vision encoder (Radford et al., 2021). To validate if comparable trends are present in *ShareLock*, we vary DINOv2-based backbones ranging from *Small* to *Giant* vision transformers (cf. Tab. 6b). Indeed, *ShareLock* also profits from scaling up the vision backbones with the average scores increasing by 33% and 11%, when moving from the *Small* to the *Base* and from the *Base* to the *Large* DINOv2 models, respectively. However, the benefits of scale start to level off thereafter, and only marginal differences are present when utilizing representations from the *Giant* vision encoder.

**Choice of projection network depth.** As the only learnable parameters, the choice of the projection networks $\mathbf{p}_{\text{txt}}$ and $\mathbf{p}_{\text{img}}$ is crucial for aligning the backbones and generalizing across downstream tasks. Table 7 reveals that projecting vision features into the language embedding space consistently underperforms compared to the reverse direction. This suggests that the vision embedding space is more semantically coherent and continuous, allowing the model to better approximate the latent concept manifold, particularly in low-data regimes. Conversely, language embeddings appear to encode non-semantic, visually irrelevant information (e.g., syntactic features), which are difficult to predict solely from visual inputs. Similarly, introducing projection layers atop both backbones and thereby removing the fixed reference space leads to reduced performance and signs of overfitting.

Table 7: **Architectural choices of projection network.** Generalization is best with sufficient transformative power (i.e., MLP layers) on top of the language backbone and direct projection into the vision space.

| $|\mathbf{p}_{\text{txt}}|$ | $|\mathbf{p}_{\text{img}}|$ | IN1k | IN-R | IN-A | Pets | Cars | Aircraft |
|---|---|---|---|---|---|---|---|
| - | 2 | 23.8 | 23.7 | 33.4 | 10.0 | 2.8 | 2.0 |
| - | 4 | 22.4 | 22.4 | 29.1 | 7.2 | 2.3 | 1.8 |
| 2 | - | 52.8 | 74.1 | 65.2 | 33.8 | 7.2 | 8.2 |
| 4 | - | 54.5 | **74.7** | **65.9** | **36.0** | 7.5 | 6.7 |
| 6 | - | **55.1** | 74.0 | 65.8 | 35.8 | **8.5** | 7.8 |
| 2 | 2 | 48.4 | 46.8 | 57.0 | 20.2 | 6.6 | **9.7** |
| 4 | 4 | 43.7 | 41.1 | 48.6 | 13.7 | 7.1 | 6.6 |

# 6 Conclusion

We systematically investigate the visual capabilities and inherent alignment of unimodal language models. Our analysis demonstrates that general LLM quality, as measured by MMLU-Pro, correlates with visual aptitude and that decoder-based models effectively generalize across modalities. By integrating off-the-shelf LLMs into a lightweight CLIP-like architecture, we leverage their large-scale pretraining, intrinsic knowledge of the visual world, and multilingual capabilities, achieving competitive performance with conventional VLMs trained on significantly larger datasets. Our findings offer a deeper understanding of state-of-the-art language models and highlight their potential for broader adoption in vision-centric applications.

## 7   Acknowledgments

The authors thankfully acknowledge the HPC resources provided by the Erlangen National High Performance Computing Center (NHR@FAU) of the Friedrich-Alexander Universität Erlangen-Nürnberg (FAU) under the BayernKI project v115be. BayernKI funding is provided by Bavarian state authorities. We thank SURF for providing GPU cluster access and support during this project. This work was supported in part by the Pioneer Centre for AI, DNRF grant number P1 and TNO's research programs Appl.AI and DSS. We gratefully acknowledge the ELLIS Unit Amsterdam for providing funding for a research visit to Copenhagen.

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

## A  Reproducibility Statement

We acknowledge and emphasize the importance of reproducibility in our work and take active measures to facilitate reproducibility efforts. Besides providing comprehensive documentation of our methods throughout the main paper, with additional details in the supplementary materials, we will publish source code for the proposed *ShareLock* model.

Our use of existing models aligns with their intended purpose and is carried out in an academic setting. Any data used follows its original access conditions, and all artifacts produced are intended for research purposes only. Only publicly available resources and scientific artifacts are used.

## B  Limitations

While we compare a wide range of publicly available encoder- and decoder-based language models, attributing concrete performance differences on mainly architectural differences is not possible due to the different pretraining objectives, training datasets, and number of parameters used in these models. As the primary focus of this work is on highlighting the visual alignment inherent to unimodal language models and their incorporation into data- and compute-efficient vision-language models, our investigation is limited to datasets with up to 12M image-caption pairs. Scaling *ShareLock* to larger datasets for further performance improvements is left for future work. Although pre-computing features only once is possible due to locked vision and language backbones, using LLMs with billions of parameters significantly increases computational costs of forward passes compared to smaller conventional encoder architectures.

## C  Acquisition of Textual Class Representations

Besides the template-based targets proposed by Radford et al. (2021) that solely substitute the respective class names, we generate more comprehensive auxiliary information about classes (e.g., Wikipedia-style articles) using the instruction-tuned version of the Llama-3-8B model and acquire human-curated information from Wikipedia (details provided in C).

Class representations are essential for facilitating the knowledge transfer between classes in the traditional definition of zero-shot learning. Compared to attributes or other forms of class semantics, language-based class representations are more conveniently accessible at various scales and may come in diverse manifestations. The advent of LLMs adds further possibilities for generating and obtaining such auxiliary information. The following paragraphs specify the respective properties and acquisition process. Here, all LLM-based class representations are generated using the instruct-tuned version of LLama-3 8B.

**Class Names.**  A set of 80 human-engineered prompt templates in the style of `"a photo of a <class name>"` are adopted from Radford et al. (2021).

**Wikipedia Page.**  Being a comprehensive and mostly factually correct source of information, Wikipedia constitutes an interesting source of auxiliary information in the context of zero-shot classification. To obtain class-article correspondences, class names are automatically matched with page names, after which additional manual quality checks are performed. Nonetheless, an ideal match does not always exist due to high class specificity or generality, in which case superordinate articles are considered or template-based fallbacks are employed.

**LLM-based Wikipedia Style Articles.**  Despite being specifically prompted for articles mimicking Wikipedia, the Llama-3-generated texts tend to show significant differences in style compared to their real counterparts.

As the lengthy nature of Wikipedia(-style) articles might dilute the information content captured by the language embeddings, the texts are split into individual sentences, which are used as targets during training. For all types of class representations, predictions are made by aggregating class scores through averaging over all individual class-specific texts.

## D  In-Depth Analysis of Visual LLM Generalization

As outlined in Section 4, we find that the general capability of language models is strongly correlated with their ability to perform well on visual tasks. While this is also true for members of the Phi-3 family of models, their absolute visual generalization scores are notably lower compared to models of similar size and capability,

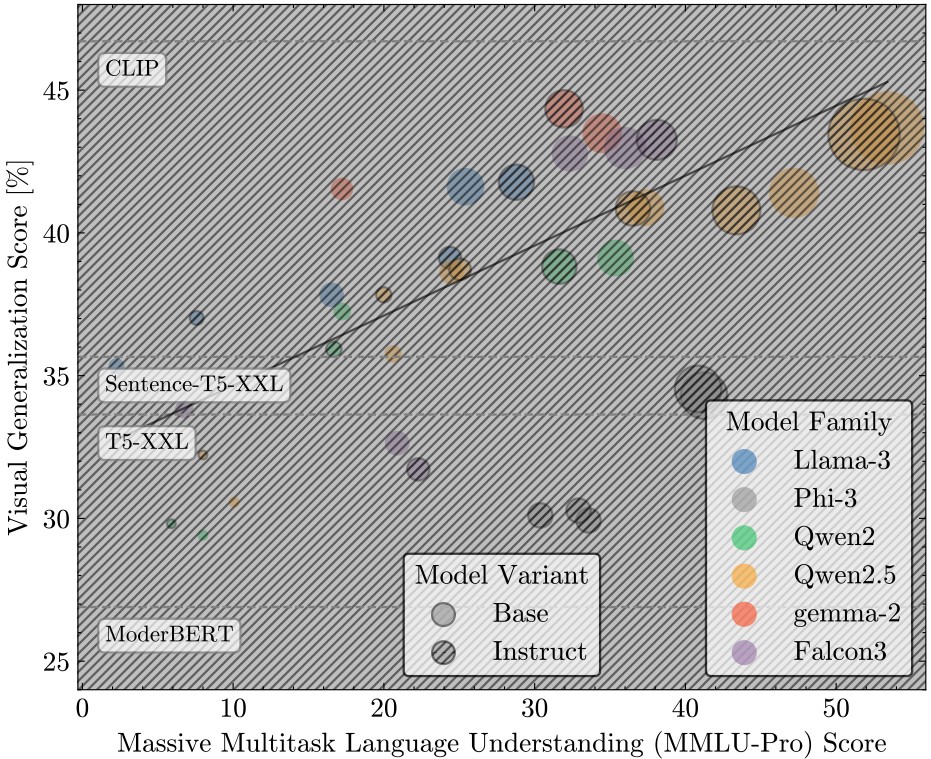

Figure 7: **Visual generalization performance relative to MMLU-Pro scores.** Model capability on language tasks is predictive of visual transfer performance of LLMs (Pearson-$r$: 0.768 and 0.523 (excl./incl. Phi-3 models)). Dot size is proportional to the LLM's parameter count.

as seen in Figure 7. This discrepancy likely illustrates the effects of the extensive data curation and synthetic data creation utilized in Phi-3, which might remove visual information to favor tokens that promote reasoning abilities. Thus, a lack of exposure to sufficient factual knowledge about real-world conditions may impede the formation of visually informed representations.

As seen in Figure 7 and Table 12, the text encoder taken from the CLIP ViT-B/16 model constitutes a strong baseline that is not yet reached by any of the current state-of-the-art LLMs. However, considering the explicit vision alignment on 400M image-text pairs makes the strong generalization abilities unsurprising. On the contrary, obtaining a score of 44.9 as the best performing unimodal model, Gemma-2 9B (Mesnard et al., 2024) is close to CLIP's 46.7 despite no multimodal exposure. This further underscores the remarkable degree of visual alignment inherent to decoder-based language models.

LLMs are often subject to additional task-specific fine-tuning. Table 8 compares different models and their derivatives tuned on instruction and multimodal data. While vision-tuned variants are available for Phi-3 (Microsoft, 2024), Qwen2 (Wang et al., 2024b), and Vicuna v1.5 (Zheng et al., 2023), XTuner's LLaVA model (XTuner Contributors, 2023) constitutes the source for the visual Llama-3 variant. Across all models, with the exception of Llama-3, the impact of fine-tuning is minor, typically shifting performance by only a few decimal points in the case of Phi-3 and Vicuna. Interestingly, Llama-3's performance declines significantly after fine-tuning ($-1.7$ and $-10.4$ for instruction and visual tuning), contrasting with the generally stable results of other models. Neither instruction-based nor visual fine-tuning shows a clear and consistent advantage in improving overall performance. These insights are also reflected in Figure 7 and Table 12. Ultimately, the base model's architecture and training regime are more significant in determining performance than post-hoc fine-tuning strategies.

Table 8: **Visual generalization ability of different fine-tuning regimes.** Fine-tuning has minimal impact on visual alignment of LLM representations. Llama-3 is a notable exception with a significant performance decrease for instruct- and vision-tuned variants.

|  | Llama-3 8B | Phi-3 Mini | Qwen2 7B | Llama-2/Vicuna 7B |
|---|---|---|---|---|
| **None** | **44.4** | n/a | 40.2 | 42.5 |
| **Instruct** | 42.7 | 36.9 | **41.8** | 42.3 |
| **Visual** | 34.0 | **37.0** | 40.3 | **42.6** |

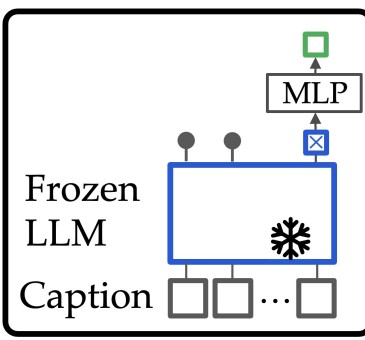

Figure 8: **Text features.** We obtain the final text features by processing the last caption token with an MLP. This allows avoiding expensive forward passes of the LLM during training by pre-computing and storing the features (×).

# E  Additional Ablations and Comparisons

## E.1  Projection Network Architecture

The multi-layer perceptron (MLP) projection networks of *ShareLock* as introduced in Section 3 are conceivably simple. As these are the only unfrozen and tunable parts of the model architecture and thus responsible for aligning vision and language inputs, they are of particular significance to aptly process and transform the inputs. Following Zhai et al. (2022), no transformation to the vision inputs is applied for any of the architectures. With a hidden size of 4096 and four layers, the MLP processing the language features comprises approximately 53M parameters.

In addition to the straightforward MLP-based networks, also more sophisticated Transformer-based architectures are inspired by recent works. First introduced as part of the BLIP-2 model (Li et al., 2023), the Q-Former is a lightweight Transformer-based model that extracts features from an input modality using cross-attention with learnable query tokens. Similarly, albeit introduced in a different context, NV-Embed (Lee et al., 2024) uses a latent attention layer to pool language tokens and receive a global embedding. Slight adjustments are made to both baseline architectures to better suit late-fusion vision-language modeling, which we will denote as *Q-Transformer* and *NV-Transformer*. While features are extracted via last-token-pooling (see Figure 8) when used with MLP projection networks, all tokens of the input sequence are considered for the alternative architectures. The hyperparameters were selected based on the implementation details suggested in the original publications and to approximately match the MLP baseline in learnable parameter count. Both the Q-Transformer and the NV-Transformer projection networks have a token dimension of 1024 in the Transformer parts of the models, eight learnable queries (Q-Transformer), and key/values (NV-Transformer). Whereas the Q-Transformer consists of 3 blocks and 4 attention heads, NV-Transformer comprises a total of four layers with eight cross-attention heads each.

The choice of projection network architecture is compared in Table 9. The evaluated models use DINOv2-ViT-B/14 as their vision backbone. While no single architecture consistently scores best, the MLP-based *ShareLock* configuration performs competitively compared to NV-Transformer and Q-Transformer throughout the evaluation datasets. Additionally, Transformer-based architectures entail increased computational complexity due to the more evolved attention mechanism and processing of more tokens, making MLPs an attractive choice from an efficiency perspective as well. These results suggest that the additional information contained across all tokens of an input is not significantly more adjuvant compared to solely considering the last token representation as is done with the MLP.

Table 9: **Comparisons of the projection network architectures tuned as part of *ShareLock* training.** Simple MLPs perform competitively compared to more advanced Transformer-based architectures.

| Architecture | IN1k | IN-R | IN-A | Pets | Cars | ESAT |
|---|---|---|---|---|---|---|
| NV-Transformer | 44.2 | 55.8 | 42.1 | 25.4 | 4.7 | 30.3 |
| Q-Transformer | 51.8 | 61.3 | 48.1 | 36.8 | **7.2** | **36.7** |
| MLP | **52.1** | **64.1** | **50.9** | **43.1** | 4.4 | 27.9 |

### E.2 Loss Function

The use of the Sigmoid Loss (SigLIP) has unlocked further efficiency and performance enhancements in the standard CLIP regime (Zhai et al., 2023). However, no substantial gains are found when using SigLIP in the *ShareLock* framework as presented in Table 10.

Table 10: **Comparison of the loss function used in *ShareLock* training.** The vanilla CLIP loss trumps the sigmoidal SigLIP loss.

| Loss Function | IN1k | IN-R | IN-A | Pets | Cars | ESAT |
|---|---|---|---|---|---|---|
| SigLIP Loss | 49.5 | 61.6 | 49.5 | 30.8 | **6.0** | **31.4** |
| CLIP Loss | **52.1** | **64.1** | **50.9** | **43.1** | 4.4 | 27.9 |

## F   Extended Visual Generalization Results

Due to limited space, the reported results on visual generalization in Table 1 and Figure 1 of the main paper are averaged accuracies across five seeds and four datasets (AWA2 (Xian et al., 2017), CUB (Wah et al., 2011), FGVCAircraft (Maji et al., 2013), and ImageNet$^+$). For increased transparency, the results are presented without dataset-level aggregation in Tables 11 and 12. Moreover, we include more detailed results corresponding to Figure 3 in Table 13. In addition to last token aggregation for decoder-based models, we also evaluate applying the same mean pooling across all tokens as utilized for encoders. However, last token pooling results in superior performance.

## G   Supplementary Qualitative Results

Figure 9 provides additional qualitative insights into the retrieval ability of CLIP, LiT, and *ShareLock* models trained on CC3M. *ShareLock* demonstrates visual understanding across a wide array of domains and levels of abstraction.

Table 11: **Visual generalization capability of various language models.** Decoder-based language models outperform encoder-based architectures across all types of input data. Llama-3 8B is used for LLM generated Wikipedia articles.

| Class Repr. | Type | Language Model | AWA2 | CUB | Aircraft | IN$^+$ | Avg |
|---|---|---|---|---|---|---|---|
| **Class Names** | **Dec** | **Llama-3 8B** | $50.1 \pm 8.3$ | $42.7 \pm 4.1$ | $29.7 \pm 5.5$ | $36.7 \pm 1.1$ | 39.8 |
| | | **Llama-3.1 8B** | $51.1 \pm 6.3$ | $44.2 \pm 2.3$ | $33.0 \pm 2.9$ | $36.3 \pm 0.8$ | 41.2 |
| | | **Gemma 7B** | $53.4 \pm 6.0$ | $42.4 \pm 3.2$ | $31.4 \pm 2.3$ | $37.0 \pm 0.6$ | 41.0 |
| | | **Gemma-2 9b** | $52.0 \pm 5.5$ | $45.7 \pm 3.7$ | $34.6 \pm 4.6$ | $39.0 \pm 1.2$ | 42.8 |
| | | **NV-Embed-v2** | $48.9 \pm 3.7$ | $42.9 \pm 4.9$ | $33.8 \pm 5.3$ | $32.2 \pm 0.4$ | 39.4 |
| | **Enc** | **T5-3b** | $47.0 \pm 7.6$ | $33.6 \pm 2.7$ | $22.6 \pm 5.7$ | $27.9 \pm 1.8$ | 32.8 |
| | | **BERT-Large** | $26.5 \pm 18.5$ | $13.2 \pm 6.5$ | $9.3 \pm 8.5$ | $7.8 \pm 2.8$ | 14.2 |
| | | **Flan-UL2** | $52.6 \pm 5.8$ | $39.4 \pm 3.1$ | $26.1 \pm 4.5$ | $33.4 \pm 0.9$ | 37.9 |
| | | **ModernBERT-embed** | $38.5 \pm 4.1$ | $31.9 \pm 3.4$ | $25.6 \pm 4.6$ | $19.6 \pm 1.7$ | 28.9 |
| | | **sentence-t5-xxl** | $60.5 \pm 6.1$ | $33.4 \pm 2.4$ | $24.1 \pm 4.3$ | $28.4 \pm 1.1$ | 36.6 |
| | | **all-roberta-v1** | $44.3 \pm 8.8$ | $31.9 \pm 1.8$ | $22.8 \pm 2.0$ | $25.1 \pm 2.3$ | 31.0 |
| **LLM Wiki Page** | **Dec** | **Llama-3 8B** | $62.4 \pm 6.6$ | $39.2 \pm 3.0$ | $37.5 \pm 3.1$ | $36.4 \pm 0.7$ | 43.9 |
| | | **Llama-3.1 8B** | $59.0 \pm 5.1$ | $41.6 \pm 3.5$ | $36.7 \pm 3.1$ | $36.5 \pm 0.6$ | 43.4 |
| | | **Gemma 7B** | $58.3 \pm 9.6$ | $41.7 \pm 2.7$ | $34.1 \pm 5.0$ | $37.1 \pm 0.8$ | 42.8 |
| | | **Gemma-2 9b** | $60.8 \pm 6.7$ | $43.1 \pm 2.6$ | $39.2 \pm 4.9$ | $36.8 \pm 0.7$ | 45.0 |
| | | **NV-Embed-v2** | $51.2 \pm 2.1$ | $47.1 \pm 1.3$ | $36.4 \pm 2.8$ | $33.5 \pm 0.6$ | 42.1 |
| | **Enc** | **T5-3b** | $48.4 \pm 9.4$ | $31.7 \pm 3.0$ | $32.4 \pm 4.1$ | $31.9 \pm 1.3$ | 36.1 |
| | | **BERT-Large** | $25.2 \pm 9.7$ | $10.4 \pm 2.5$ | $16.7 \pm 4.6$ | $13.2 \pm 2.3$ | 16.4 |
| | | **Flan-UL2** | $57.7 \pm 12.0$ | $36.3 \pm 3.0$ | $34.3 \pm 3.7$ | $35.6 \pm 0.6$ | 41.0 |
| | | **ModernBERT-embed** | $43.1 \pm 5.9$ | $22.6 \pm 2.6$ | $9.3 \pm 4.8$ | $20.6 \pm 0.6$ | 23.9 |
| | | **sentence-t5-xxl** | $58.0 \pm 5.6$ | $37.4 \pm 2.0$ | $28.7 \pm 5.4$ | $32.1 \pm 2.6$ | 39.1 |
| | | **all-roberta-v1** | $46.2 \pm 6.5$ | $36.9 \pm 1.4$ | $29.4 \pm 7.2$ | $25.3 \pm 0.8$ | 34.5 |
| **Wikipedia** | **Dec** | **Llama-3 8B** | $62.4 \pm 7.8$ | $41.5 \pm 2.1$ | $41.3 \pm 3.2$ | $32.8 \pm 0.5$ | 44.5 |
| | | **Llama-3.1 8B** | $57.8 \pm 7.8$ | $40.2 \pm 2.8$ | $40.7 \pm 3.0$ | $32.6 \pm 0.4$ | 42.8 |
| | | **Gemma 7B** | $60.3 \pm 4.6$ | $41.5 \pm 3.3$ | $39.2 \pm 5.6$ | $33.0 \pm 0.3$ | 43.5 |
| | | **Gemma-2 9b** | $56.2 \pm 8.4$ | $43.5 \pm 3.2$ | $44.3 \pm 3.9$ | $34.0 \pm 0.5$ | 44.5 |
| | | **NV-Embed-v2** | $58.2 \pm 5.7$ | $47.9 \pm 2.1$ | $35.6 \pm 3.5$ | $32.5 \pm 0.4$ | 43.6 |
| | **Enc** | **T5-3b** | $60.4 \pm 9.3$ | $34.8 \pm 3.8$ | $34.2 \pm 3.9$ | $29.0 \pm 1.6$ | 39.6 |
| | | **BERT-Large** | $47.0 \pm 16.3$ | $8.0 \pm 2.7$ | $27.7 \pm 6.1$ | $13.1 \pm 1.7$ | 24.0 |
| | | **Flan-UL2** | $64.5 \pm 9.2$ | $42.9 \pm 3.1$ | $35.9 \pm 3.9$ | $32.5 \pm 0.5$ | 44.0 |
| | | **ModernBERT-embed** | $37.6 \pm 4.7$ | $21.5 \pm 2.6$ | $15.4 \pm 6.4$ | $18.3 \pm 0.6$ | 23.2 |
| | | **sentence-t5-xxl** | $63.4 \pm 4.7$ | $43.0 \pm 2.4$ | $33.4 \pm 6.7$ | $31.4 \pm 2.2$ | 42.8 |
| | | **all-roberta-v1** | $53.3 \pm 5.0$ | $39.0 \pm 2.8$ | $32.8 \pm 4.6$ | $31.8 \pm 0.5$ | 39.23 |

| Language Model | AWA2 | CUB | Aircraft | IN$^+$ | Avg |
|---|---|---|---|---|---|
| **CLIP-B/16 Text Encoder** | $63.0 \pm 5.2$ | $51.5 \pm 2.2$ | $37.1 \pm 3.4$ | $35.3 \pm 0.6$ | 46.7 |
| **Qwen2-0.5B** | $40.8 \pm 4.1$ | $27.2 \pm 1.6$ | $25.6 \pm 3.5$ | $24.1 \pm 0.6$ | 29.4 |
| **Qwen2-0.5B-Instruct** | $40.9 \pm 3.4$ | $27.8 \pm 2.0$ | $26.7 \pm 3.3$ | $23.9 \pm 0.6$ | 29.8 |
| **Qwen2-1.5B** | $52.8 \pm 6.0$ | $33.9 \pm 1.7$ | $31.7 \pm 6.1$ | $30.6 \pm 0.4$ | 37.2 |
| **Qwen2-1.5B-Instruct** | $48.1 \pm 5.9$ | $33.6 \pm 1.4$ | $32.1 \pm 6.7$ | $29.9 \pm 0.8$ | 35.9 |
| **Qwen2-7B** | $52.3 \pm 5.8$ | $38.6 \pm 2.5$ | $29.6 \pm 5.3$ | $35.9 \pm 0.7$ | 39.1 |
| **Qwen2-7B-Instruct** | $50.9 \pm 3.2$ | $40.1 \pm 3.9$ | $28.9 \pm 5.8$ | $35.3 \pm 0.8$ | 38.8 |
| **Qwen2.5-0.5B** | $49.0 \pm 4.5$ | $27.9 \pm 5.2$ | $20.8 \pm 8.0$ | $24.5 \pm 1.2$ | 30.6 |
| **Qwen2.5-0.5B-Instruct** | $49.0 \pm 3.8$ | $31.4 \pm 3.9$ | $24.2 \pm 4.3$ | $24.3 \pm 1.3$ | 32.2 |
| **Qwen2.5-1.5B** | $55.5 \pm 7.8$ | $35.9 \pm 3.6$ | $22.1 \pm 7.1$ | $29.6 \pm 0.9$ | 35.8 |
| **Qwen2.5-1.5B-Instruct** | $58.3 \pm 9.8$ | $37.2 \pm 2.7$ | $26.6 \pm 7.8$ | $29.3 \pm 0.8$ | 37.8 |
| **Qwen2.5-14B** | $47.8 \pm 4.7$ | $47.4 \pm 1.8$ | $33.8 \pm 5.4$ | $36.6 \pm 0.5$ | 41.4 |
| **Qwen2.5-14B-Instruct** | $47.3 \pm 6.2$ | $46.7 \pm 3.4$ | $32.5 \pm 3.2$ | $36.7 \pm 0.5$ | 40.8 |
| **Qwen2.5-32B** | $55.3 \pm 4.7$ | $47.8 \pm 2.4$ | $33.7 \pm 4.5$ | $37.9 \pm 0.6$ | 43.7 |
| **Qwen2.5-32B-Instruct** | $54.9 \pm 4.0$ | $47.0 \pm 1.3$ | $33.9 \pm 5.0$ | $38.1 \pm 0.7$ | 43.4 |
| **Qwen2.5-3B** | $52.4 \pm 7.1$ | $39.5 \pm 4.1$ | $30.7 \pm 5.4$ | $32.0 \pm 0.7$ | 38.6 |
| **Qwen2.5-3B-Instruct** | $54.8 \pm 9.8$ | $37.9 \pm 5.0$ | $30.7 \pm 3.4$ | $31.4 \pm 0.5$ | 38.7 |
| **Qwen2.5-7B** | $51.3 \pm 2.2$ | $41.7 \pm 3.4$ | $35.6 \pm 2.6$ | $35.0 \pm 0.8$ | 40.9 |
| **Qwen2.5-7B-Instruct** | $53.4 \pm 2.1$ | $40.4 \pm 3.2$ | $34.9 \pm 2.4$ | $34.8 \pm 0.7$ | 40.9 |
| **gemma-2-2b** | $54.9 \pm 4.2$ | $40.0 \pm 3.2$ | $27.3 \pm 5.4$ | $34.3 \pm 0.8$ | 39.1 |
| **gemma-2-2b-it** | $60.7 \pm 7.6$ | $39.5 \pm 5.3$ | $31.4 \pm 4.0$ | $34.5 \pm 0.6$ | 41.5 |
| **gemma-2-9b** | $54.5 \pm 3.6$ | $46.5 \pm 2.6$ | $35.6 \pm 6.2$ | $37.4 \pm 0.4$ | 43.5 |
| **gemma-2-9b-it** | $59.6 \pm 6.3$ | $44.1 \pm 2.9$ | $37.9 \pm 1.2$ | $35.7 \pm 0.5$ | 44.3 |
| **Meta-Llama-3.2-1B** | $49.0 \pm 5.7$ | $31.8 \pm 2.1$ | $29.2 \pm 4.5$ | $31.3 \pm 0.7$ | 35.3 |
| **Meta-Llama-3.2-1B-Instruct** | $53.7 \pm 8.3$ | $35.4 \pm 3.6$ | $27.9 \pm 3.7$ | $31.1 \pm 0.7$ | 37.0 |
| **Meta-Llama-3.2-3B** | $46.2 \pm 7.2$ | $37.9 \pm 1.8$ | $32.2 \pm 3.1$ | $35.0 \pm 0.7$ | 37.8 |
| **Meta-Llama-3.2-3B-Instruct** | $53.8 \pm 10.9$ | $35.7 \pm 2.6$ | $33.0 \pm 4.8$ | $34.0 \pm 0.7$ | 39.1 |
| **Meta-Llama-3-8B-Instruct** | $52.7 \pm 6.4$ | $42.0 \pm 3.7$ | $36.8 \pm 5.8$ | $35.6 \pm 0.6$ | 41.8 |
| **Meta-Llama-3.1-8B** | $51.1 \pm 4.5$ | $45.4 \pm 1.9$ | $33.8 \pm 3.2$ | $36.2 \pm 0.7$ | 41.6 |
| **Phi-3-medium-128k-instruct** | $38.6 \pm 5.5$ | $36.1 \pm 1.8$ | $30.4 \pm 2.8$ | $32.1 \pm 0.4$ | 34.3 |
| **Phi-3-medium-4k-instruct** | $38.2 \pm 5.6$ | $36.0 \pm 3.5$ | $31.3 \pm 3.0$ | $32.6 \pm 0.4$ | 34.5 |
| **Phi-3-mini-128k-instruct** | $36.2 \pm 6.1$ | $28.7 \pm 2.8$ | $27.5 \pm 4.2$ | $27.9 \pm 1.2$ | 30.1 |
| **Phi-3-mini-4k-instruct** | $35.9 \pm 5.8$ | $28.5 \pm 3.5$ | $27.9 \pm 5.0$ | $27.4 \pm 1.8$ | 29.9 |
| **Phi-3.5-mini-instruct** | $37.0 \pm 5.4$ | $30.1 \pm 1.8$ | $27.1 \pm 4.5$ | $27.0 \pm 2.5$ | 30.3 |
| **Falcon3-10B-Base** | $60.0 \pm 6.8$ | $44.6 \pm 1.5$ | $32.7 \pm 2.8$ | $34.6 \pm 0.4$ | 43.0 |
| **Falcon3-10B-Instruct** | $59.0 \pm 7.4$ | $46.0 \pm 1.7$ | $32.8 \pm 5.7$ | $35.3 \pm 0.5$ | 43.3 |
| **Falcon3-1B-Base** | $50.0 \pm 8.0$ | $33.1 \pm 1.4$ | $23.3 \pm 5.8$ | $28.9 \pm 0.7$ | 33.8 |
| **Falcon3-3B-Base** | $48.7 \pm 5.6$ | $30.9 \pm 2.5$ | $21.9 \pm 9.7$ | $29.1 \pm 0.8$ | 32.6 |
| **Falcon3-3B-Instruct** | $47.2 \pm 4.2$ | $29.3 \pm 3.0$ | $20.9 \pm 12.0$ | $29.4 \pm 0.7$ | 31.7 |
| **Falcon3-7B-Base** | $56.9 \pm 8.2$ | $44.0 \pm 1.9$ | $35.1 \pm 4.2$ | $35.1 \pm 0.5$ | 42.8 |

Table 12: **Visual generalization performance across language models.** Larger and more capable models within a family facilitate better generalization in visual tasks and thus boast increased visual alignment

Table 13: **Decoder vs. encoder language models.** Using the Ettin model family (Weller et al., 2025), we compare the effects of pretraining objectives on visual alignment with consistent training data and model sizes. In these experiments, decoder-based models consistently outperform encoders.

| # Params. [M] | Language Model | AWA2 | CUB | FGVCAircraft | ImageNet$^+$ | Average |
|---|---|---|---|---|---|---|
| **17** | **Decoder (Avg)** | 24.2 | 31.2 | 16.0 | 12.3 | 20.9 |
| | **Decoder (Last)** | 35.7 | 29.7 | 10.6 | 12.6 | 22.1 |
| | **Encoder (Avg)** | 20.6 | 27.7 | 16.7 | 12.4 | 19.4 |
| **32** | **Decoder (Avg)** | 26.7 | 29.4 | 16.0 | 14.2 | 21.6 |
| | **Decoder (Last)** | 36.5 | 24.3 | 9.9 | 14.6 | 21.3 |
| | **Encoder (Avg)** | 20.4 | 30.9 | 16.5 | 13.8 | 20.4 |
| **68** | **Decoder (Avg)** | 32.0 | 27.6 | 19.9 | 16.9 | 24.1 |
| | **Decoder (Last)** | 37.5 | 21.5 | 20.7 | 17.8 | 24.4 |
| | **Encoder (Avg)** | 21.1 | 30.2 | 18.5 | 14.7 | 21.1 |
| **150** | **Decoder (Avg)** | 38.3 | 29.0 | 16.6 | 19.3 | 25.8 |
| | **Decoder (Last)** | 39.8 | 26.5 | 24.4 | 20.6 | 27.8 |
| | **Encoder (Avg)** | 33.1 | 30.5 | 16.3 | 17.8 | 24.5 |
| **400** | **Decoder (Avg)** | 36.9 | 32.1 | 21.3 | 21.7 | 28.0 |
| | **Decoder (Last)** | 37.4 | 28.2 | 26.4 | 24.3 | 29.1 |
| | **Encoder (Avg)** | 35.3 | 30.0 | 17.8 | 19.6 | 25.7 |
| **1000** | **Decoder (Avg)** | 48.4 | 32.8 | 21.9 | 25.5 | 32.1 |
| | **Decoder (Last)** | 46.8 | 31.7 | 20.4 | 29.3 | 32.1 |
| | **Encoder (Avg)** | 41.8 | 32.3 | 19.2 | 25.3 | 29.6 |

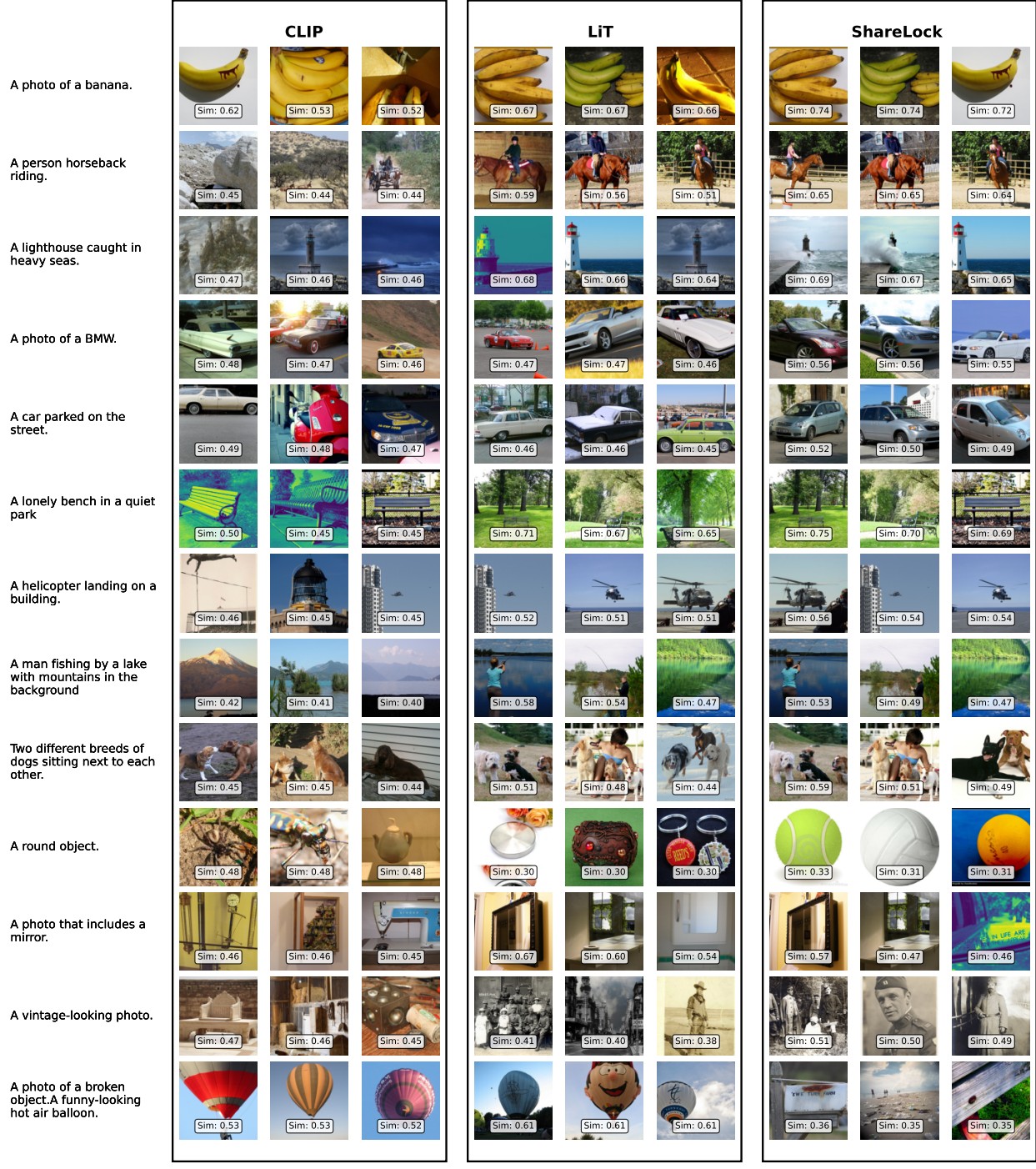

Figure 9: **Qualitative comparison on text-to-image retrieval (ImageNet-1k).**

