# OpenReview forum: "Better Language Models Exhibit Higher Visual Alignment"
_TMLR — Accepted by TMLR_

### Review · Reviewer_rso2 · 2025-11-10

**Summary Of Contributions:**

Main Contributions
Direct Assessment of Visual Alignment in LLMs
The paper provides the first rigorous evaluation of how LLMs' internal text representations correspond to visual concepts, using strict zero-shot image classification tasks. This avoids concept leakage and tests true generalization.
Decoder-Based LLMs Excel at Visual Tasks
Modern decoder-based LLMs (e.g., Llama-3, Gemma-2) outperform encoder-based models (e.g., BERT) in visual generalization. The ability of an LLM to perform well on language tasks (measured by MMLU-Pro) strongly predicts its visual alignment and transfer performance.
ShareLock: Efficient Vision-Language Model
The authors introduce ShareLock, a lightweight VLM that combines frozen LLM and vision model features. ShareLock achieves strong multimodal capabilities, including:
Robust zero-shot classification
Cross-lingual generalization (e.g., 38.7% accuracy for Chinese vs. CLIP’s 1.4%)
Competitive performance with models trained on much larger paired datasets
Reduced Need for Paired Data and Compute
By leveraging frozen LLMs, the approach significantly reduces the need for large-scale paired image-text data and compute resources, making VLMs more accessible and adaptable.
Multilingual and Compositional Reasoning
ShareLock demonstrates strong cross-lingual transfer and above-CLIP performance in compositional reasoning tasks, highlighting the rich semantic content of LLM representations.

Key Findings
LLMs Encode Visual Knowledge: Even without multimodal training, LLMs contain semantically meaningful representations that can be mapped to visual latent spaces.
Model Size and Capability Matter: Larger and more capable LLMs show higher visual alignment and generalization.
Minimal Fine-Tuning Needed: The intrinsic visual alignment of LLMs is robust to fine-tuning, and the base model’s architecture and training regime are more important than post-hoc strategies.
Efficiency: ShareLock achieves competitive results with orders of magnitude less data and compute compared to conventional VLMs like CLIP.

Why It Matters
For researchers and practitioners: This work shows that advances in LLMs can directly benefit vision-language tasks, even without expensive multimodal training.
For multilingual and low-resource settings: The approach enables strong performance in languages and domains with limited paired data.
For future VLM development: The findings suggest that leveraging frozen LLMs is a promising direction for building efficient, robust, and generalizable multimodal models.

**Additional Comments:**

N/A

**Audience:**

Yes

**Audience Explanation:**

suit for low computing resource researchers, since it seems only linear layer probation were involved in the work.

**Broader Impact Concerns:**

Even though I don't think the idea of paper is quite innovative, but I want to emphasize that this paper provide researchers who are interested in LLM and are limited with computing resource some insight about how to do LLM related research within limited capacity.

**Claims And Evidence:**

Yes

**Claims Explanation:**

Key Evidences:

1. Zero-Shot Image Classification Benchmarks
The authors rigorously evaluate multiple LLMs (both encoder- and decoder-based) on strict zero-shot image classification tasks across four diverse datasets (AWA2, CUB, FGVCAircraft, ImageNet+).
They enforce strict disjointness between training and test classes to avoid concept leakage, which is a common pitfall in multimodal evaluation.

2. Comparative Performance Tables
Detailed tables show that decoder-based LLMs (e.g., Gemma-2, Llama-3) consistently outperform encoder-based models (e.g., BERT, RoBERTa) in visual generalization scores.
The correlation between language model capability (MMLU-Pro scores) and visual alignment is quantified (Pearson r = 0.768), supporting the claim that better LLMs yield better visual transfer.

3. ShareLock Model Results
ShareLock, which uses frozen LLM and vision model features, is benchmarked against established models like CLIP and LiT.
It achieves competitive or superior performance in zero-shot classification, especially in low-data regimes, and demonstrates strong cross-lingual generalization (e.g., 38.7% accuracy for Chinese vs. CLIP’s 1.4%).

4. Ablation Studies
The paper includes ablations on:
Language model choice (decoder vs. encoder)
Vision backbone choice
Projection network architecture (MLP vs. Transformer)
Loss function (CLIP loss vs. SigLIP)
These studies reinforce the robustness and generality of the main findings.

5. Qualitative Results
The authors provide qualitative retrieval examples showing ShareLock’s ability to align text and images across fine-grained and abstract prompts.

6. Limitations and Transparency
The paper openly discusses limitations, such as the inability to attribute performance differences solely to architecture due to varying pretraining data and objectives.
It also notes the scale of datasets used and computational constraints, which adds credibility to the claims.


Areas for Further Validation
Scaling to Larger Datasets: The authors acknowledge that their experiments are limited to datasets with up to 12M image-caption pairs. Further validation on even larger, more diverse datasets would strengthen the generalizability of the findings.
Compositional Reasoning: While ShareLock improves over CLIP, it still falls short of human-level performance in compositional reasoning tasks, suggesting room for future work.

Conclusion
Overall, the paper provides strong and multi-faceted evidence for its main arguments.
The combination of quantitative benchmarks, ablation studies, and qualitative results makes the claims credible and well-supported. The authors are transparent about limitations, which further strengthens the trustworthiness of their conclusions.

**Requested Changes:**

In the paper, they claim that decoder-only LLM performs better than encoder-only ones in vision-alignment tasks. However most of encoder-only models listed in the paper seems developed much earlier before decoder-only one, in terms of publish date.  It would be good there is some recent dp with encoder-only model to compare.

---

> ### Author Response · Authors · 2025-11-21
>
> We thank reviewer rso2 for their review and respond to their points in the following:
>
> ---
>
> *"... most of encoder-only models listed in the paper seems developed much earlier before decoder-only one, in terms of publish date. It would be good there is some recent dp with encoder-only model to compare."*
>
> In the submitted version of the paper, we compare against ModernBERT (2024) and Flan-UL2 (2024) which were released around the same time as the decoder-based models we compare them to. With the SentenceT5 and Flan-UL2 models we consider encoders of comparable parameter sizes with popular LLMs. However, recent developments in the NLP community have focused heavily on generative LLMs (incl. for representation learning: Lee et al., 2024; Springer et al.,2024) and conventional encoder-based models trained at comparable scales are rare. We would like to point the reviewer to Figure 3, where we compare models trained on the same data and with the number of parameters but using masked language modeling and next token prediction objectives, respectively. This family of models was released in 2025 and allows for a fair comparison and statements about decoder-based approaches outperforming encoder-based ones.
>
> ---
>
> *"Even though I don't think the idea of paper is quite innovative, but I want to emphasize that this paper provide researchers who are interested in LLM and are limited with computing resource some insight about how to do LLM related research within limited capacity."*
>
> We thank the reviewer for highlighting the compute and data efficiency of our lightweight alignment approach. However, we argue that our study also contributes the following important insights and innovations:
>   - We introduce a novel and rigorous evaluation benchmark to assess the visual alignment of language representations and find that visual generalization performance is highly correlated with language comprehension (MMLU-Pro) for LLMs.
>   - Interestingly, we identify that decoder-based LLMs are particularly effective at representing visual concepts, making us the first to present a systematic study on effects of the pretraining objective on visual alignment.
>   - We incorporate these findings into the lightweight ShareLock VLM, leveraging pretrained models with minimal paired alignment training to achieve results competitive with models trained with orders of magnitude more data or compute.
>   - In addition to general vision-language performance, we showcase that LLM abilities like multilingual understanding transfer into ShareLock without explicit training (38.7% IN1k performance in Chinese vs. CLIP’s 1.4% despite small-scale English-only training data).

---

### Review · Reviewer_ori9 · 2025-11-14

**Summary Of Contributions:**

The paper aims to investigate how well large language models understand visual data.

 However, as explained below, I am unable to fully comprehend the paper because the narrative is convoluted, overly parenthetical, and filled with details that are meaningful only to readers who are already highly knowledgeable in this area (may even only to this solution).

**Audience:**

Yes

**Audience Explanation:**

As said, my feeling is that, if the paper would solve the clarity issue, would be fine. But I would like to really pass an opinion on this matter, after the paper is clear. Emphasize  "If" and "feeling" not "sure" and "certainty".

**Broader Impact Concerns:**

Not for the moment. Again, the main problem is that I was not able to understand the details of the paper. Once I manage to do this, the "Impact concern" aspect needs to be re-evaluated.

**Claims And Evidence:**

No

**Claims Explanation:**

My feeling is that the paper might be fine, if explained properly. I have not found a detail that is wrong. The topic and findings suggest that they are interesting.

The problem is that in the current form the paper is very hard to read, until it lost me. While I am not the most suitable reader for this paper, yet I am in the TMLR auditorium, so I should not get lost in detail, parenthesis, cross -references, etc. I should not need to draw additional schematics just to be keep track of which is what!

In my view, the paper fails to meed the following TMLR criteria: "clarity of the narrative".

**Requested Changes:**

The main issues is that the paper is, in general, very difficult to follow. I have not identified incorrect content, but the paper appears to be written for an audience of experts who already understand the core concepts. As a result, it focuses almost exclusively on details. Additionally, the writing style requires significant polishing.

More specifically:

**Introduction / Section 3**:
- Page 1: The paragraph beginning with “To ensure a rigorous … in language representation” seems unnecessary here. These details belong in the implementation section, not the introduction.
- Figure 1:
   - What is the source of this figure?
   - What is MMLU-Pro? (I eventually found it, but a non-expert would need to stop reading, look it up, and return.)
   - What is the “visual generalization score”? (Same issue.)
   - More importantly, a figure placed in the introduction should motivate the paper, but this one does not clearly communicate the motivation.
- The statement “We observe non-trivial zero-shot generalization across language model types, indicating latent visual-semantic alignment.” What generalization should the reader be seeing in the figure? And why does this justify the stated conclusion?
- The sentence “In particular, features from modern generative LLMs outperform classic encoder-based embeddings such as BERT … This advantage of decoder-based models persists …” The first sentence discusses generative LLM features, but the second refers to “the advantage of the decoder.” Where is this decoder-based advantage actually shown? ("This" implies that it was described in the previous sentence, but it is not!)

- Include a high-level schematic illustrating the core experiment: how LLMs are used for visual alignment. Provide a simple example.
- Add one or more schematics explaining key components and design choices:
   - the LLM,
   - embedding extraction,
  - the VLM encoder–decoder setup, etc.

- Explain in more detail where zero-shot classification occurs and how it fits into the workflow.
- Section 3 (“Method”) begins abruptly with detailed discussion of zero-shot classification and train/test splits. Provide more context before going into these details.
- Table 1: What is its origin? What should the reader understand from it?
- Pg3 :"their similarity is computed as the cosine similarity, given by the dot product of the embeddings", The similarity is between Z_txt and Z_img?
- Page 5: The sentence “A new insight resulting from our analysis is the competitiveness of decoder-based language models in representing visual concepts.” This is the first time the paper suggests that evaluating “the competitiveness of decoder-based language models” is a core objective. Please clarify this purpose earlier.

**Experiments**
- Table 2: Clarify in the caption that accuracy is the evaluation metric.
- ShareLock: The paper does not explain what this method is. Searching for the term leads to mentions in the introduction, the caption of Figure 3, and the “experimental setup → methodology” section, but nowhere is there a clear description. Provide a dedicated explanation.
- The sentence “As illustrated in Figure 3, this design places the LLM at the core of the model and necessitates expressive language representations for strong VLM performance.”
 - There is no LLM visible in Figure 3. Perhaps it is labeled as “text”? But given the complexity of the method, the reader should not have to guess. Also, what in Figure 3 demonstrates that expressive language representations are necessary?
- “We increase the projection network  $p_{text}(⋅) $ to four layers with a hidden dimension of 4096.” The earlier definition of $p_{text}$  is easy to miss, requiring the reader to stop and search for it. The explanation is overburdened with parenthetical details.
- Then: “The additional parameters … enhance ShareLock’s ability to capture complex, non-linear cross-modal correspondences.”
Which parameters? Presumably the weights of the additional MLP layers, but this should be stated clearly.
-Next: “Additionally, the maximum number of steps is increased to 5000.” Steps of what?  Optimization steps? Again, unclear.

- Clarify what the “conventional CLIP-like methods” used for comparison are, why they are relevant, and how they differ from the proposed approach.
- In general, the method section is too difficult to follow. It needs a clearer, more structured explanation of the problem, the architecture, and the reasoning behind the design choices.




**Citation**
-When rewriting, please take into account the journal’s citation style. The manuscript has been obviously composed for short numerical references (e.g., [17]), . Yet the journal uses a full name style that disrupts readability. Adapt the text to the journal’s required format and avoid overloading sentences that present the method  with citations. Place citations in a separate sentence.

I could continue with similar remarks for the **rest of the manuscript**, but ultimately it is not my role to rewrite the entire paper. The presentation needs to be to substantially revised,  to improve clarity, structure, and readability.

---

> ### Author Response · Authors · 2025-11-21
> **Author's response to reviewer comments (1/2)**
>
> We thank reviewer ori9 for their extensive feedback on our submission based on which we have made the manuscript more accessible to readers from different domains. We clarify and answer the reviewer’s questions in the following and are happy to elaborate and discuss further.
>
> ---
>
> *"The paragraph beginning with “To ensure a rigorous … in language representation” seems unnecessary here. These details belong in the implementation section"*
>
> Small modifications to improve the readability of the introduction have been made in the updated submission. The specific excerpt pointed to by the reviewer (strict separation of concepts in training and evaluation) is not just a mere technical detail but imperative to how we propose to assess the “true” visual generalization and alignment performance of language representations. Hence, we argue that it is warranted and important to introduce the reader to this unique and important aspect about our evaluation protocol early. While it is first introduced on a high-level in the introduction, all specifics about the dataset splits are found in Section 3.3.
>
> ---
>
> *Clarification of source and content of Figure 1*
>
> Figure 1 is created based on our experimental exploration where we measure the visual alignment across various different language models. The advantage of decoder-based models is apparent from their higher VGS scores compared to encoder-based baselines (dashed horizontal lines). We updated the caption and content of Figure 1 to improve clarity. Moreover, Figure 1 is described in detail in Section 4, where we introduce more background on MMLU-Pro metric and identify important takeaways from the Figure.
>
> ---
>
> *"The sentence “In particular, features from modern generative LLMs outperform classic encoder-based embeddings such as BERT … This advantage of decoder-based models persists …” The first sentence discusses generative LLM features, but the second refers to “the advantage of the decoder.” Where is this decoder-based advantage actually shown? ("This" implies that it was described in the previous sentence, but it is not!)"*
>
> As indicated in the snippet highlighted by the reviewer, the first sentence establishes that “LLMs outperform classic encoder-based embeddings”. That is what the “advantage” held by LLMs that the second sentence refers to. Generative LLMs are decoder-based models and these terms are used synonymously throughout the paper.
>
> ---
>
> *"Include a high-level schematic illustrating the core experiment" + "Add one or more schematics explaining key components and design choices"*
>
> Based on the reviewer’s feedback, we introduced a new schematic (Figure 2) that illustrates the two different kinds of data used for VLM training and alignment probing and details how contrastive training and inference with zero-shot classification is performed. This complements the previously included Figures 4 (ShareLock architecture compared to prior works; used to be Figure 3) and 8 (extraction of text representations from LLMs; used to be Figure 7).
>
> ---
>
> *"Section 3 (“Method”) begins abruptly with detailed discussion of zero-shot classification and train/test splits. Provide more context before going into these details."*
>
> We thank the reviewer for this suggestion and implemented it accordingly.
>
> ---
>
> *"Table 1: What is its origin? What should the reader understand from it?"*
>
> Table 1 summarizes the visual generalization performance of various language models for different types of class representations as indicated in the “LLM representations encode visual knowledge.” paragraph that first refers to this table. The reported visual generalization scores were obtained by following the protocol specified in Section 3.3. As mentioned by the second sentence of the caption (third in updated submission), the main takeaway is the competitiveness of decoder-based language representations. We improved the clarity of the caption accordingly.
>
> ---
>
> *"Pg3 :"their similarity is computed as the cosine similarity, given by the dot product of the embeddings", The similarity is between Z_txt and Z_img?"*
>
> The reviewer is correct and the similarity is between z_txt and z_img. We add an explicit mention of this to the sentence in question.
>
> ---
>
> *"... the paper suggests that evaluating “the competitiveness of decoder-based language models” is a core objective. Please clarify this purpose earlier."*
>
> The competitiveness of decoder-based models is itself not a core objective per se, but rather a novel insight resulting from our rigorous study of different language model types that breaks with common practice as prior works predominantly incorporate encoder-based models into VLM pipelines. We highlight this finding as such in the abstract, introduction, and summary of contributions.
>
> ---
>
> *"Table 2: Clarify in the caption that accuracy is the evaluation metric."*
>
> We thank the reviewer for highlighting this point and made according updates to the caption.
>
> ---

---

> ### Author Response · Authors · 2025-11-21
> **Author's response to reviewer comments (2/2)**
>
> *"ShareLock: The paper does not explain what this method is."*
>
> The reviewer is correct that the definition of ShareLock in Section 5.1 builds upon the architecture and training setup introduced in Section 3. With the addition of Figure 2, we illustrate the differences in data between the alignment probing and VLM training modes more clearly. Figure 4 correspondingly provides a high-level schematic of ShareLock’s architecture and compares it to prior CLIP-like models. We made changes to improve the clarity and independence of the introduction of ShareLock while maintaining curtness and without introducing intrusive redundancies.
>
> ---
>
> *"There is no LLM visible in Figure 3."*
>
> We added a legend to easily distinguish between encoders (trapezoids) and decoder-based embedding methods (rectangle) based on the reviewer’s suggestion.
>
> ---
> *"“We increase the projection network $p_{text}(⋅) $ to four layers with a hidden dimension of 4096.” The earlier definition of $p_{text}$ is easy to miss, requiring the reader to stop and search for it. The explanation is overburdened with parenthetical details."*
>
> We address the reviewer’s concern and introduce additional context to make the function and application of the projection network clearer.
>
> ---
>
> *"“The additional parameters … enhance ShareLock’s ability to capture complex, non-linear cross-modal correspondences.” Which parameters? -Next: “Additionally, the maximum number of steps is increased to 5000.” Steps of what? Optimization steps? Again, unclear."*
>
> The “additional parameters” refer to the weights of the projection network that we increase from a simple linear layer (alignment probing) to a 4-layer MLP (ShareLock) as described in the previous sentence (increase in layers → additional parameters). The mentioned steps are the number of optimization steps. We clarified this accordingly.
>
> ---
>
> *"Clarify what the “conventional CLIP-like methods” used for comparison are, why they are relevant, and how they differ from the proposed approach."*
>
> In the following, we expand on the type of baselines we compare to.
>
> Concretely, Fig. 2 (left) illustrates the standard CLIP-style dual encoder in which both the image and text backbones are trained from scratch with a contrastive loss. This setup is used by OpenAI CLIP, SLIP, LaCLIP, and the DataComp CLIP variants we compare to. These methods reflect the dominant recipe for late-fusion VLM models and are trained on datasets of various scales.
>
> Fig. 2 (center) corresponds to LiT, which keeps the vision encoder frozen but still trains the text encoder and projection layers with the same contrastive objective. LiT therefore sits between CLIP and our method by already exploiting a strong pretrained vision backbone, but still requiring fine-tuning the text encoder.
>
> Fig. 2 (right) depicts ShareLock, which goes one step further by freezing both the vision and language encoders and only learning lightweight projection networks on top of their pretrained features. This “fully frozen” configuration is what enables us to precompute features once and train with very low compute while retaining competitive performance.
>
> All of these three methodologies we refer to as “CLIP-like” as they all share the same dual-encoder late fusion architecture and are optimized with the same contrastive alignment objective.
>
> Additionally, ASIF shares the idea of frozen unimodal encoders but is training-free, using a closed-form alignment instead of optimizing a projection head. We therefore include it as a complementary baseline at the extreme end of the efficiency spectrum.
>
> Taken together, the selected baselines trace a spectrum from fully trainable CLIP-style models (high flexibility but high data/compute requirements), through partially frozen LiT, to our fully frozen ShareLock and the training-free ASIF. Comparing along this spectrum isolates the effect of progressively freezing more of the model while keeping the overall architecture and objective comparable.
>
> In the revised submission, we more explicitly state the connection to Figure 2 in Sec. 5.1 and clarify which methods fall into each configuration.

---

### Review · Reviewer_hAtV · 2025-11-19

**Summary Of Contributions:**

**Contributions**
1. The authors evaluate the visual alignment inherent to language models using probing.
2. The authors find that decoder-based LLMs are effective sources of visual knowledge, better than encoder-based models.
3. The authors propose ShareLock, a frame work to integrate LLMs into lightweight VLM and achieve robust generalization results.

**Strength**
1. The paper structure is clear and easy to follow.
2. The experiments and analysis are comprehensive. The design of ShareLock is simple yet effective.
3. The findings that decoder-based LLMs are more effective than encoder-based models in visual generalization is interesting.

**Weakness**
1. In experimental setup 3.3, the authors claim that one variant of DINOv2 models are used. However, it is very likely that this visual backbone have been trained on the seen classes data, even though in a self-supervised manner. This contradicts the motivation of "enforcing disjointness between seen and unseen concepts" on page 1.
2. The "inherent visual alignment" (section 5, page 6) is not precise. To show the "inherent", the ideal way is to illustrate a projection layer on non-LLM text representations perform much worse. Comparing among LLMs does not show there is inherent alignment (table 1, section 4).
3. For experiments in section 5, the baselines are using different visual backbones from ShareLock (CLIP v.s. DINOv2). At least it is not clear from "Baselines and comparisons" section. If the visual backbones are not the same, the comparison is not fair.
4. Some figures and tables lack clarity. For example, in Figure 1, it is not clear how Visual Generalization score is computed. In Figure 3, there is no LLM but in text the authors describe "As illustrated in Figure 3, this design places the LLM at the core of the model ...". In Table 5, it is not clear what BERT-base and Llama-3 8B represents.

**Audience:**

Yes

**Audience Explanation:**

This paper provides insights to the researchers working on vision and language alignment.

**Claims And Evidence:**

No

**Claims Explanation:**

See weakness 1, 2 and 3. The major concern is in experimental settings:

1. It is not clear whether pre-trained DINOv2 has seen *unseen class* data or not.
2. The "inherent visual alignment" is not precise from the resultsof only LLM-based models in Table 1 or Figure 1.
3. For ShareLock experiments in section 5, it seems the visual backbones are not consistent.

**Requested Changes:**

1. Please clarify or address weaknesses.
2. It would be better if the authors could provide a computational cost analysis for experiments in section 5. For example, the inference time and FLOPs for different methods.

Other minor suggestions:
For "seen and unseen concepts" in page1, the reader would easily get confused as there is no explanation.

---

> ### Author Response · Authors · 2025-11-21
> **Author's response to reviewer comments**
>
> We thank reviewer hAtV for their feedback on our submission. We will address their points in the following.
>
> ---
>
> *"It is not clear whether pre-trained DINOv2 has seen unseen class data or not."*
>
> As indicated by the reviewer, it needs to be assumed that concepts classified as “unseen” during our alignment training were included in the broad pretraining of DINOv2. However, in the context of our work, the terms “seen” and “unseen” apply to the concepts on which the language model was explicitly aligned to the vision model.
> Strict disjointness still holds as we align and measure the generalization of the model with non-overlapping sets of classes. Moreover, familiarity with all concepts through exposure during pretraining is an advantage as it prevents confounding unimodal out-of-domain generalization with the degree of cross-modal alignment that we intend to measure.
> Based on the reviewer’s feedback, we replaced the terminology of “seen” and “unseen” concepts adopted from the literature with “aligned” and “unaligned” which better capture the methodology followed in our work. Additionally, we elaborate more on this aspect in the updated submission and illustrate the process visually by introducing Figure 2.
>
> ---
>
> *"The "inherent visual alignment" is not precise from the results"*
>
> We agree with the reviewer that experiments on separate non-language-based targets can be helpful to illustrate how the visual alignment is inherent to LLMs. Additional results on non-LLM representations are presented in the following table.
>
> | Class Representation |  Visual Generalization Score (VGS)  |
> |----------------------|:-----:|
> | One-Hot Encoding     |  4.5% |
> | word2vec             | 25.5% |
> | GloVe                | 28.0% |
> | fastText             | 25.5% |
> | Llama-3 (8B)         | 39.8% |
> | Gemma-2 (9B)         | 42.8% |
>
> As a naive language-free representation, we encode the classes via one-hot vectors. As one might expect, this results in a near-random VGS of 4.5% as models have no semantic continuity connecting seen and unseen representations. Better generalization is facilitated by employing static word embedding models like word2vec, GloVe, or fastText. However, while these simple embedding methods are more semantic than one-hot encodings, LLMs exhibit significantly higher VGSs. This illustrates that, because of the design of our benchmark that strictly distinguishes aligned and unaligned concepts, the continuity and semantics of the class representations are critical for generalization to novel concepts. The VGS is thus determined by the degree of visual alignment of the features extracted from language features (i.e., their inherent alignment). We will include these new results in Section 4.
>
> ---
>
> *"For ShareLock experiments, it seems the visual backbones are not consistent."*
>
> The reviewer is correct that different visual backbones are used in CLIP vs. our ShareLock method. This is because the CLIP recipe learns the vision encoder from scratch whereas the ShareLock methodology leverages pretrained models. However, the LiT baseline uses the exact same vision features as ShareLock (DINOv2-ViT-L/14) to ensure a fair comparison. This is mentioned in the “Baselines and comparisons” paragraph of Section 5.1 but we revised the paragraph a bit to make this point clearer.
>
> ---
>
> *"It would be better if the authors could provide a computational cost analysis for experiments in section 5. For example, the inference time and FLOPs for different methods."*
>
> The LLMs utilized by ShareLock are typically significantly bigger than the text encoders used by CLIP and SigLIP, making forward passes longer by about 5.5x for Llama 3 (8B) compared to BERT-Base (0.1B). However, for CLIP-style zero shot image classifications, the language targets only need to be precomputed once for the set of classes to distinguish between. Therefore, the actual real-world inference performance is only dependent on the used vision encoder. Consequently, CLIP, LiT and our ShareLock method are roughly equivalent in terms of FLOPs and inference time as long as the same backbone architecture is used. For the DINOv2-based ViT-L/14 encoder used as the backbone in ShareLock and LiT, this equates to around 160 GFLOPS for a single-image forward pass at 224x224 resolution. We added this aspect to Section 5.1.
> During training, our model benefits greatly from precomputable features and only one forward pass through the LLM is required. Concrete estimates of our model’s training efficiency are included in Section 5.1.
>
> ---
>
> *"Some figures and tables lack clarity."*
>
> We address this with the following improvements:
>  - Expanding on the origin and meaning of the VGS and MMLU-Pro scores in the caption of the Figure 1 itself.
>  - Adding a legend to easily distinguish between encoders (trapezoids) and decoder-based embedding methods (rectangle).
>  - Indicating that BERT-base and Llama-3 are the (frozen) language backbones utilized for each method.

---

### Review · Reviewer_7e5z · 2025-11-23

**Summary Of Contributions:**

- This paper explores whether stronger large language models exhibit better alignment with vision representations.
- They proposed Sharelock, a simple framework that freezes both a pretrained vision encoder and a decoder-only LLM and only trains a projection from text space.
- With this setup, the authors show that decoder-only LLMs achieve higher visual alignment and zero-shot classification accuracy than the traditional encoder-based text models, even with small paired datasets.
- The work further established a positive correlation between LLM language proficiency and cross-model generalization performance, suggesting that linguistic capacity transfers to visual grounding.
- The paper also highlights strong multilingual transfer capabilities, achieving notable improvements on non-English zero-shot classification tasks without additional training.
- Key strengths include a clean and controlled experimental design, extensive comparisons across text models, and a practical low-compute recipe for vision-language alignment.

**Additional Comments:**

This is a clean, empirical paper with thoughtful controls and clear takeaways. It solidifies an emerging trend that high-capacity LLMs, even without fine-tuning, possess semantic benefits for visual grounding. The paper's simplicity is both its strength and its limitations. The compositional and reasoning gaps are not explored. With improved transparency around compute and additional diagnostics, and with sound reasoning, it will be beneficial.

**Audience:**

Yes

**Audience Explanation:**

- Yes, the paper would be interesting to a broader segment of the TMLR audience, particularly those studying vision-language models, V-L alignment, representation learning, and the integration of the large language models into perception tasks.
- The finding that decoder LLMs inherently encode more visually grounded semantics than the traditional text encoders offers a simple but powerful insight for people developing data-efficient multimodal systems.
- The multilingual and low-data generalization results also have practical relevance for resource-constrained and cross-lingual applications.

**Broader Impact Concerns:**

- The work poses minimal ethical risks in itself, as it trains lightweight alignment layers on frozen, publicly available models.
- Improved multilingual alignment could amplify existing biases in LLM text representations when projected into vision space.
- The author could strengthen the broader impact section by briefly analyzing fairness implications and potential downstream misuse.

**Claims And Evidence:**

Yes

**Claims Explanation:**

Claims

1. Decoder LLMs align better to a frozen vision space than encoder text models.
2. A small text-side head on frozen backbones yields strong zero-shot results in low-data regimes.
3. Language proficiency correlated with visual generalization.

- The evidence presented in the paper is clear and convincing.
- The authors conduct well-controlled experiments that isolate the role of the text encoder by freezing both the vision and language backbones.
- The positive correlation between language ability and visual alignment is consistently observed across various models and datasets.
- Ablation on projection depth, data size, and multilingual transfer further support the main claims.
- However, some aspects are overstated. The reported efficiency (“under one GPU-hour”) excludes the heavy cost of LLM feature extraction.
- Also, the compositional benchmarks reveal limitations, which they attributed to the architecture choice of LLMs. Despite this, the empirical evidence substantiates the primary findings that decoder-based LLMs provide more transferable visual alignment than prior encoder text models.

**Requested Changes:**

1. Clarification of the compute claims, either abstract or methodology, can be revised to report the full cost, including LLM feature precomputation, rather than only the projection training time.
2. Need for stronger compositional evaluation, include other compositional benchmarks like ARO, SugarCrepe. To support the claims about the visual-semantic alignment and explain the reasoning limitations over compositionality.
3. Details about how multilingual label translations were normalized.
4. Include encoder-only models matched in parameter count and context window to ensure that the decoder outperforms encoders; the conclusion is not confounded.
5. Add a detailed discussion on why decoder-based language knowledge fails on compositional reasoning tasks.

---

> ### Author Response · Authors · 2025-11-25
> **Author's response to reviewer comments (1/2)**
>
> We thank reviewer 7e5z for their feedback on our submission and will address their points in the following.
>
> ---
>
> *1. Clarification of the compute claims to include full cost (incl. precomputation)*
>
>
> The reviewer correctly points out that only the actual training is considered in the abstract for brevity. Section 5.1 provides full estimates for ShareLock, including precomputation of features. In addition to roughly one GPU hour of training, an additional one and eight hours are required to extract vision and language features, respectively. Our recently updated submission additionally expands on the inference costs associated with our method. Despite using language backbones with 5.5x longer forward passes, inference cost in zero-shot classification settings are bound by the vision encoder as language targets only have to be computed once, setting ShareLock on par with CLIP and LiT models.
>
> *2. & 5. Discussion on why decoder-based language knowledge fails on compositional reasoning tasks (5.) and need for stronger compositional evaluation, include other compositional benchmarks like ARO, SugarCrepe. (2.)*
>
> We agree with the reviewer that studying the limitations of CLIP-like models for compositionality is interesting and that ARO and SugarCrepe offer perspectives complementary to Winoground. However, we found that using strong frozen language features alone only gives marginal improvements and that the primary factors holding back compositional reasoning in contrastive VLMs persist in our setting.
>
> Joint embedding vision–language models (i.e., CLIP, LiT, ShareLock) struggle on linguistically fine-grained tasks and tend to collapse captions with only nuanced differences into similar representations, reducing the ability to accurately discern between detailed referential expressions or attributes. Recent works by Alhamoud et al. (2025) and Koishigarina et al. (2025), and Gurung et al. (2025) identified the contrastive alignment and particularly the conventional image-caption data as the root-cause of poor CLIP performance. A lack of fine-grained hard negatives, low attribute density in captions, and the global representation objective leads to coarse alignment and bag-of-words behavior. Koishigarina et al. further argue that, even though the necessary information might be encoded in the individual modalities, insufficient cross-modal alignment and architectural limitations prevent a more fine-grained understanding. As our method project directly into the manifold of a frozen vision encoder, another limiting factor impeding systematic consideration of fine-grained details is that the visual embedding space might not support object-level or relational grounding (Zhang et al., 2024; Jose et al., 2024).
>
> In our paper and with the inclusion of results on Winoground, we intend to show that ShareLock performs marginally better than alternatives, but that compositional reasoning capabilities remain severely limited. Since our models and baselines are trained with conventional image-caption data without adding (synthetic) hard negatives or architectural improvements aimed at improving compositionality, we do not expect novel insights from conceptually related benchmarks. For compositionality, approaches as the ones outlined above have to be followed. We will add this discussion to the final version of the paper.
>
> *3. "Details about how multilingual label translations were normalized."*
>
> Multilingual classification was performed following the same procedure as all other experiments and no special or additional processing steps were involved. The only difference is that - instead of English class names and templates - translated versions in other languages were fed into the language model. We use the translated versions from CLIP Benchmark (Cherti et al.). The extracted language representations were then L2-normalized to compute their cosine similarities to the image features. Hence, changes are only made in the input (string) space.
>
> *4."Include encoder-only models matched in parameter count and context window to ensure that the decoder outperforms encoders; the conclusion is not confounded."*
>
> With the SentenceT5 and Flan-UL2 model variants, we consider encoders of comparable parameter sizes with popular LLMs. However, recent developments in the NLP community have focused heavily on generative LLMs (incl. for representation learning: Lee et al., 2024; Springer et al., 2024) and conventional encoder-based models trained at comparable scales are rare. For this reason, we included the analysis shown in Figure 3 and the corresponding paragraph in Section 4 where we compare models trained on the same data and with an equal number of parameters but using masked language modeling and next token prediction objectives, respectively. These experiments allow for a fair comparison and statements about decoder-based approaches outperforming encoder-based ones, as all other potentially confounding factors are controlled for.

---

> ### Author Response · Authors · 2025-11-25
> **Author's response to reviewer comments (2/2)**
>
> *Broader Impact Concerns*
>
> We agree with the reviewer that the potential adverse effects of aligning open-weight models with our proposed methodology are minimal. However, the reviewer raises an important point about encoding or amplifying existing biases. Indeed, cross-linguistic studies show that large text corpora in many languages encode similar social and semantic biases, e.g. gender stereotypes (e.g., occupational or attribute associations) are observed across languages such as English, Spanish, German, and Chinese (e.g. Bolukbasi et al., 2016). There are bias directions that are shown across languages (e.g. Caliskan et al., 2017), e.g. racial, socioeconomic, and affective biases. These biases can be reflected in downstream models even when no explicit “harmful” objective is present.
>
> There is a realistic risk that vision–language alignment will inherit or amplify such biases, especially because many widely used image–caption datasets are drawn predominantly from Western, higher-income contexts (e.g., COCO, CC3M).
> At the same time, recent work such as MetaCLIP 2 (Chuang et al., 2025) explicitly tackles this by scaling CLIP training to worldwide, multilingual web data to improve coverage across languages and cultures. Our work is complementary: instead of scaling data and compute, we show that frozen LLMs can already provide strong cross-lingual transfer without task-specific multilingual alignment data or model fine-tuning.
>
> To more concretely assess robustness and fairness, we additionally evaluate our models on the Dollar Street dataset (Gaviria Rojas et al., 2022), which assesses performance across geographic and socioeconomic diversity using images of everyday objects from around the world.
>
>
> | Training Dataset   | Model     | Acc@5 |
> |--------------------|-----------|------:|
> | CC3M               | CLIP      |  49.5 |
> | CC3M               | LiT       |  75.2 |
> | CC3M               | ShareLock |  77.6 |
> | CC12M              | CLIP      |  65.3 |
> | CC12M              | LiT       |  79.6 |
> | CC12M              | ShareLock |  80.0 |
> | Proprietary (400M) | CLIP      |  80.3 |
>
> Despite being trained on substantially fewer and less diverse samples than OpenAI’s CLIP (400M proprietary pairs), ShareLock attains comparable performance on Dollar Street, while strongly improving over small-scale CLIP models. This suggests that ShareLock can leverage the broader conceptual coverage of its unimodal backbones to achieve more robust and fairer performance, even when the alignment stage itself uses limited web data.
>
> Our ShareLock method offers a practical path to robust alignment using publicly available models, modest paired data, and low compute. This makes it more easily accessible to low-resource communities. Because it may inherit biases from its pretrained components, use in sensitive domains should include domain-specific fairness checks and mitigation. We will add a Broader Impact Section to reflect these considerations.

---

### Decision · Action_Editor_qGUa · 2026-01-03

**Recommendation:** Accept as is

**Audience:**

Yes

**Audience Explanation:**

All reviewers think the paper will be of interest to TMLR audience. After considering the paper, the reviews, and the rebuttals, AE concurs with the reviewers on this.

**Claims And Evidence:**

Yes

**Claims Explanation:**

All reviewers think the claims made in the submission were supported by accurate, convincing and clear evidence. After reviewing the paper and the discussion between reviewers and authors, AE can confirm it is the case.